# The *Plasmodium* liver-specific protein 2 (LISP2) is an early marker of liver stage development

Devendra Kumar Gupta[1,2†], Laurent Dembele[2,3†],
Annemarie Voorberg-van der Wel[4], Guglielmo Roma[5], Andy Yip[2],
Vorada Chuenchob[6], Niwat Kangwanrangsan[7], Tomoko Ishino[8],
Ashley M Vaughan[6], Stefan H Kappe[6], Erika L Flannery[6],
Jetsumon Sattabongkot[9], Sebastian Mikolajczak[1,6], Pablo Bifani[2,10,11],
Clemens HM Kocken[4], Thierry Tidiane Diagana[1,2]*

[1]Novartis Institute for Tropical Diseases, Emeryville, United States; [2]Novartis Institute for Tropical Diseases, Singapore, Singapore; [3]Faculty of Pharmacy, Université des Sciences, des Techniques et des Technologies de Bamako (USTTB), MRTC – DEAP, Bamako, Mali; [4]Department of Parasitology, Biomedical Primate Research Centre, Rijswijk, Netherlands; [5]Novartis Institutes for BioMedical Research, Basel, Switzerland; [6]Center for Infectious Disease Research, Seattle, United States; [7]Faculty of Science, Mahidol University, Bangkok, Thailand; [8]Graduate School of Medicine, Ehime University, Toon, Japan; [9]Faculty of Tropical Medicine, Mahidol Vivax Research Center, Bangkok, Thailand; [10]Singapore Immunology Network (SIgN), Singapore, Singapore; [11]Department of Microbiology and Immunology, Yong Loo Lin School of Medicine, National University of Singapore, Singapore, Singapore

*For correspondence:
thierry.diagana@novartis.com

†These authors contributed equally to this work

**Abstract** *Plasmodium vivax* hypnozoites persist in the liver, cause malaria relapse and represent a major challenge to malaria elimination. Our previous transcriptomic study provided a novel molecular framework to enhance our understanding of the hypnozoite biology (Voorberg-van der Wel A, et al., 2017). In this dataset, we identified and characterized the Liver-Specific Protein 2 (LISP2) protein as an early molecular marker of liver stage development. Immunofluorescence analysis of hepatocytes infected with relapsing malaria parasites, in vitro (*P. cynomolgi*) and in vivo (*P. vivax*), reveals that LISP2 expression discriminates between dormant hypnozoites and early developing parasites. We further demonstrate that prophylactic drugs selectively kill all LISP2-positive parasites, while LISP2-negative hypnozoites are only sensitive to anti-relapse drug tafenoquine. Our results provide novel biological insights in the initiation of liver stage schizogony and an early marker suitable for the development of drug discovery assays predictive of anti-relapse activity.
DOI: https://doi.org/10.7554/eLife.43362.001

## Introduction

*Plasmodium vivax* is the second most prevalent malarial pathogen, with a wider geographical distribution than *P. falciparum*, suggested to be a risk of malaria infection for 2.5 billion people (*Howes et al., 2016*). According to the WHO report (2017), an estimated 8.5 million new clinical cases of *P. vivax* was reported in 2016 globally. Despite its high prevalence in many malaria endemic countries, *P. vivax* research is restricted to few laboratories and limited progress has been made

(*Armistead and Adams, 2018*). Notwithstanding, the FDA recently approved tafenoquine as a radical cure therapy and prophylactic for *P. vivax* malaria infection (*Frampton, 2018*). This is a significant advance as tafenoquine is administered as a single dose regimen, which is a very important improvement for patient compliance when compared to the lengthy 14-day drug regimen of its closely related predecessor primaquine. However, tafenoquine is only approved for patients over the age of 16 and, like primaquine, it cannot be administered to patients who have glucose-6-phosphate dehydrogenase (G6PD) deficiency, a common genetic disorder in malaria endemic countries, due to serious adverse side-effects and life-threatening drug-induced hemolysis (*Wells et al., 2010*; *Mazier et al., 2009*). Therefore, new drugs are critically needed to enable malaria elimination.

Malaria transmission begins when uni-nucleated sporozoites are transmitted by mosquito bite, reach the liver and invade hepatocytes within which they transform into multi-nucleated hepatic schizonts. Mature schizonts release merozoites that infect red blood cells (RBCs) and lead to the onset of clinical symptoms associated with malaria. Remarkably, sporozoites of *P. vivax, P. ovale* and *P. cynomolgi* can generate latent forms known as hypnozoites (*Prudêncio et al., 2011*). Hypnozoites, triggered by unknown signals, periodically activate several weeks (or even months) after the initial infection to cause malaria relapse (*Wells et al., 2010*; *Shanks and White, 2013*). Activation of hypnozoites was suggested to be responsible for 90% of the global clinical burden associated with relapsing malaria (*Adekunle et al., 2015*). Despite recent advances in development of models to study hepatic relapses in vitro (*Dembélé et al., 2014*; *Gural et al., 2018*; *Roth et al., 2018*) and in vivo (*Mikolajczak et al., 2015*; *March et al., 2013*), the quest for novel radical cure therapies is stymied by our poor understanding of the molecular determinants of hypnozoite persistence and activation.

The simian relapsing malaria parasite *P. cynomolgi (Pcy)*—closely related to the *P. vivax* human malaria parasite—has been crucial to our current understanding of the hypnozoite biology (*Dembélé et al., 2014*; *Krotoski et al., 1982*; *Cogswell, 1992*; *Voorberg-van der Wel et al., 2017*) and the discovery of novel liver-stage active compounds (*Zeeman et al., 2014*; *Zeeman et al., 2016*) and anti-relapse drug candidates (*Campo et al., 2015*). Here, using in vitro and in vivo liver stage infection models, we demonstrated that LISP2 is a molecular marker of initiation of liver-stage development in relapsing malaria parasites. As proof of concept, using known drugs that have differential effect on hypnozoites, we showed that the LISP2 marker could assess anti-relapse drug activities in vitro.

## Results

### LISP2 is an early marker of parasite development in the liver

To increase our understanding of the molecular mechanisms that may be underlying the physiological regulation of hypnozoite persistence and activation, we and others have recently carried out comparative transcriptomics analysis of persistent, non-replicating and replicating liver-stages of the *P. cynomolgi* parasite (*Voorberg-van der Wel et al., 2017*; *Cubi et al., 2017*). Searching for potential markers specific to dormancy in our transcriptomic dataset has proven to be difficult because hypnozoites appear to globally suppress transcription and very few genes are specifically up-regulated in hypnozoites (*Voorberg-van der Wel et al., 2017*). We reasoned that markers of hypnozoite activation and schizogony might be easier to identify as we speculated that they would be significantly enriched in large replicating parasites albeit detectable at least in a fraction of smaller parasites that might be activating hypnozoites or earliest developing forms. Looking for genes with such an expression pattern, we looked for genes that were detected above >10 FPKM (Fragments Per Kilobase Per Million) in hypnozoite (727 genes) and that were in the top 5% genes expressed in schizonts (134 genes). Amongst the seven genes fulfilling these criteria (most of them were encoding hypothetical proteins; see *Supplementary file 1*), we identified the gene PcyM_0307500, the *P. cynomolgi* ortholog of the rodent *Plasmodium berghei* Liver-Specific Protein 2 (LISP2) (*Orito et al., 2013*). In *P. cynomolgi*, LISP2 is dramatically upregulated (over 72-fold, p-value 0.0009) in replicating schizonts but, nonetheless, detectable at significant level (average FPKM 12) in hypnozoites. A similar expression pattern was reported for the *P. vivax* LISP2 (PVP01_0304700) in the recently published

liver stage transcriptome (*Gural et al., 2018*). PVP01_0304700 was found to be highly expressed in (>150, Transcripts per million, TPM) mix population (mixture of all hepatic stages), whereas it was significantly detected (>25 TPM) in hypnozoite enriched samples above threshold (25 TPM).

LISP2 protein is 2038 amino-acids (aa) and 2519 aa long in *P. cynomolgi* and *P. vivax*, with an N-terminal signal peptide and a conserved C-terminal 6-cysteine domain (6-cys) found in a family of apicomplexan-specific proteins which are mostly surface proteins (*Arredondo and Kappe, 2017*; *Thompson et al., 2001*). To determine the subcellular localization of LISP2 in relapsing malaria parasites, we raised antibodies against the conserved 6-cys domain for both predicted proteins encoded by the *P. cynomolgi* (PcyM_0307500) and *P. vivax* (PVP01_0304700/PVX000975) LISP2 genes (*Figure 1A*). We confirmed the specificity of both *P. cynomolgi* and *P. vivax* LISP2 antibodies against the recombinant antigen by western blot (*Figure 1—figure supplement 1A and B*). Cultivated simian hepatocytes infected with *P. cynomolgi* B-strain sporozoites (spz) were then analyzed in immuno-fluorescence assay (IFA) with anti-LISP2 and anti-HSP70 antibodies to monitor LISP2 expression over time. *P. cynomolgi* infected simian hepatocytes from three different macaques were maintained in vitro from day 1 to day 12 in monolayer (*Figure 1B*) and switched to the previously reported long term in vitro sandwich culture model (*Dembélé et al., 2014*) from day 13 to day 21 (*Figure 1—figure supplement 2*). LISP2 immunostaining was only observed starting at day 3 in a characteristic crescent-shape asymmetrically located on one side of the early developing liver parasites, or trophozoites (LISP2+). By day 4, LISP2 protein is distributed all around the peripheral membrane in a circular pattern around the parasites that were all very clearly multinucleated (>3 nuclei). By day 5 in larger schizonts, LISP2 appears to accumulate in peripheral vacuolar membrane structures that were not labeled by DAPI or HSP70 antibody (LISP2++). We observed those vacuoles from day 6 onwards, as the parasites mature and grow to larger schizonts. During this intensive parasite growth phase (Day 4 to Day 10), LISP2 staining was observed in the same vacuolar and peripheral membrane pattern (LISP2++) in multinucleated parasites (*Figure 1B*). Although the resolution of DAPI staining does not allow for an accurate count of the number of nuclei, LISP2+ parasites generally appear to have few nuclei (see representative images on *Figure 1—figure supplement 3*). The HSP70 protein is detected during all the *Plasmodium* liver life-cycle stages and HSP70 immunostaining is seen for all parasites from day 1 and throughout the entire 21 days period. We independently confirmed this expression pattern at the RNA level in Fluorescent In Situ Hybridization (FISH) experiments in *P. cynomolgi* cultures. Indeed, the LISP2 transcript is robustly expressed in schizonts but more weakly in a subset of smaller parasites (*Figure 1—figure supplement 4*).

## Assessment of LISP2 expression in *P. vivax* liver stages

We next determined the LISP2 expression in vivo in human liver chimeric mice (huHep mice) infected with the *P. vivax* human parasite (*Mikolajczak et al., 2015*). The huHep mice were inoculated with 1 $\times 10^6$ of *P. vivax* sporozoites. *P. vivax* infected mice were sacrificed at days 3, 5, 8 and 18 and liver tissues were examined for the presence of hepatic stages by performing IFA. Liver stages were assessed using *P. vivax* antibodies against LISP2 and UIS4 (upregulated in infectious sporozoite 4) proteins. Similar to the in vitro observations with *P. cynomolgi* parasites, LISP2 expression is first observed at days 3 post-infection in *P. vivax* with an identical crescent staining located at one side of the membrane. During the course of development from day 3 to day 18, we observed a large fraction of non-replicating small parasites (hypnozoites) that were devoid of LISP2 expression in *P. vivax*. At each time point (days 3, 5, 8 and 18), LISP2- hypnozoite and LISP2+ trophozoite were of similar size ranges. As reported earlier, UIS4 protein accumulated in a characteristic prominence in established mature hypnozoites (*Mikolajczak et al., 2015*) as well as in LISP2+ trophozoites while in mature schizonts the entire parasitophorous vacuole membrane (PVM) were stained (*Figure 2A*). Interestingly, we also observed that UIS4 and LISP2 proteins co-localized at the PVM prominence in LISP2+ trophozoites at all time-points (*Figure 2B*). We could not perform UIS4 and LISP2 co-staining in *P. cynomolgi* as both antibodies were made in same species, notwithstanding we could demonstrate the presence of UIS4 PVM prominence in established hypnozoites in *P. cynomolgi* (*Figure 2—figure supplement 1*). Collectively, the above data from in vitro and in vivo suggests that LISP2 is expressed in early developing liver stage parasites and represents a marker for initiation of liver stage schizogony in relapsing malaria species.

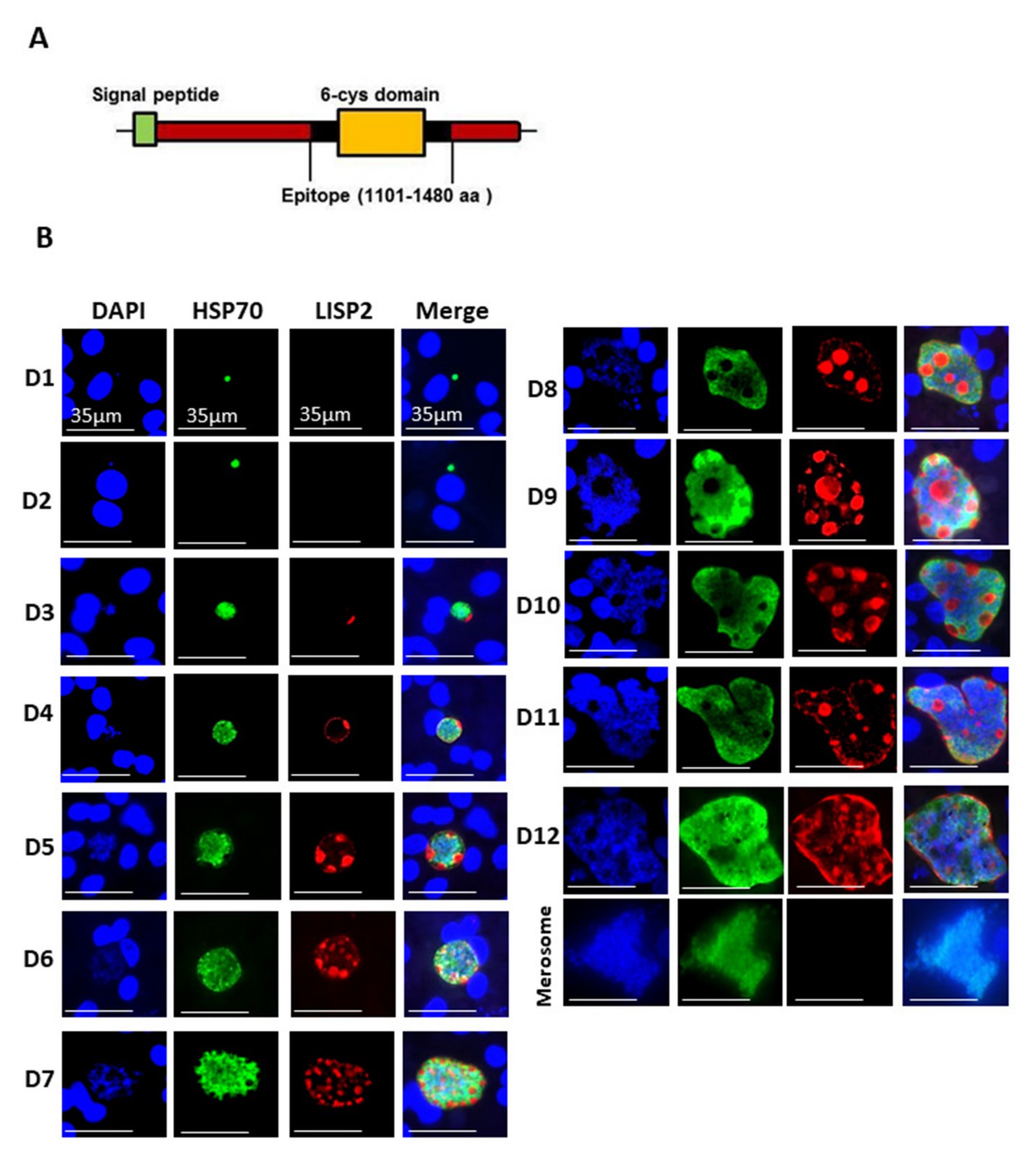

**Figure 1.** LISP2 marks the beginning of hepatic stage development. (**A**) Schematic structure of the LISP2 gene showing the N-terminal signal peptide, 6-cys domain and epitope (highlighted in black) used as immunogen to raise antibodies against *P. cynomolgi* and *P. vivax* LISP2 protein (**B**) Immunofluorescence assay, IFA of *P. cynomolgi* liver stage parasites in infected simian primary hepatocytes showing liver stage development from day 1 to day 12 post-infection visualized with DAPI for DNA content (blue), a polyclonal antibody specific for *P. cynomolgi* HSP70 protein (green) and a
*Figure 1 continued on next page*

*Figure 1 continued*

monoclonal antibody specific for *P. cynomolgi* LISP2 protein (red); Scale bar, 35 µm. From day 3 onwards, LISP2 staining observed in a crescent shape asymmetrically located on one side of earliest developing parasites (LISP2+). By day 5, LISP2 accumulates in peripheral vacuolar membrane structures in multinucleate schizonts and does not co-localize with DAPI or HSP70 (LISP2++).

DOI: https://doi.org/10.7554/eLife.43362.002

The following figure supplements are available for figure 1:

**Figure supplement 1.** Recognition of recombinant LISP2 antigen with purified anti-LISP2 monoclonal antibody by western blot.

DOI: https://doi.org/10.7554/eLife.43362.003

**Figure supplement 2.** LISP2 expression in liver stages beyond 13 days post-sporozoite infection.

DOI: https://doi.org/10.7554/eLife.43362.004

**Figure supplement 3.** LISP2+ parasites on Day 3 post-infection.

DOI: https://doi.org/10.7554/eLife.43362.005

**Figure supplement 4.** LISP2 fluorescent in situ hybridization (FISH) staining of *P. cynomolgi* liver stages parasites.

DOI: https://doi.org/10.7554/eLife.43362.006

**Figure supplement 5.** Validation of specific LISP2 RNA-FISH signal.

DOI: https://doi.org/10.7554/eLife.43362.007

## LISP2 expression differentiates dormant hypnozoites from developing liver stages

Because LISP2 expression was first detected in day 3 in early developing parasites of small size, we reasoned that LISP2 antibodies might distinguish the persistent hypnozoites population from those similarly sized earliest developing trophozoites. To visualize and quantify the different hepatic stages using LISP2 expression, we first carried out LISP2 immunostaining analysis at day 6 post-infection. Our results revealed the presence of three distinct populations of liver stage parasites. In total, 23% of the day 6 liver stage parasites robustly expressed the LISP2 protein (LISP2++), whereas the vast majority of the liver stage parasites (73%) remained smaller, uni-nucleated, and did not react with the LISP2 monoclonal antibodies (LISP2-). In addition to these large multinucleated parasites, we also observed a small fraction of LISP2 positive trophozoites (4%) with a characteristic crescent-shape LISP2 staining pattern (LISP2+) (*Figure 3A*).

Next, we set out to quantify the parasite growth kinetics and daily measured the liver stage size for each of these three parasite populations (LISP2-, LISP2+ and LISP2++) (*Figure 3B*). In the first 8–9 days, persistent LISP2- hypnozoites increase in size from about 1 µm diameter to reach a plateau at about 6 µm diameter on average. Similarly, persistent *P. vivax* hypnozoites have been reported to undergo a slight growth phase *in vivo* (*Mikolajczak et al., 2015*). In sharp contrast, the size of LISP2+ trophozoites does not change over time and remained constant from day 3 to day 21 at a median size of 6.9 µm (5th to 95th percentile; 6.3 and 7.3) in diameter (*Figure 3B*). Thus, remarkably, from day 7 to day 8 onwards, LISP2- hypnozoites and LISP2+ trophozoites were indistinguishable by size. As expected, developing LISP2++ liver stage schizonts were significantly bigger at day 4 with a median diameter of 9.4 µm (5th to 95th percentile; 6.9 and 15.1) and continue to grow to reach a plateau by day 10 with a median size of about 21.6 µm (5th to 95th percentile; 9.1 and 49) in diameter for mature schizonts. Collectively, the LISP2 expression pattern suggested that LISP2+ parasites may be appearing continuously throughout the culture period and develop into LISP2++ maturing schizonts ranging from 7 to 8 µm up to 49 µm in diameter.

We next wanted to quantify the fraction of parasites observed daily for each of the three populations (LISP2-, LISP2+ and LISP2++) over the 21-day observation period. The results of experiments carried out with three independent batches of *P. cynomolgi* sporozoites and three independent sources of simian hepatocytes are reported in *Figure 3—figure supplement 1*. Using LISP2 and HSP70 antibodies, we monitored the proportion of different hepatic forms of the parasite from day 1 to day 21 (*Figure 3—figure supplement 1A and B*). The total number of parasites counted in each day for 3 hepatocyte donors is reported in the graphs in *Figure 3—figure supplement 1C*. As expected the total number of parasites decreased over time as they grew to mature schizonts and merosomes. In the first 2 days following the sporozoite infection, we detected only small LISP2- parasites positively stained with the HSP70 antibody. These LISP2- uni-nucleated parasites persisted as hypnozoites until day 21 and constitute the majority (≥65%) of the parasite liver stages from day 3 onwards. A small fraction (≤10%) of LISP2+ trophozoites first appeared on day 3 to reach a

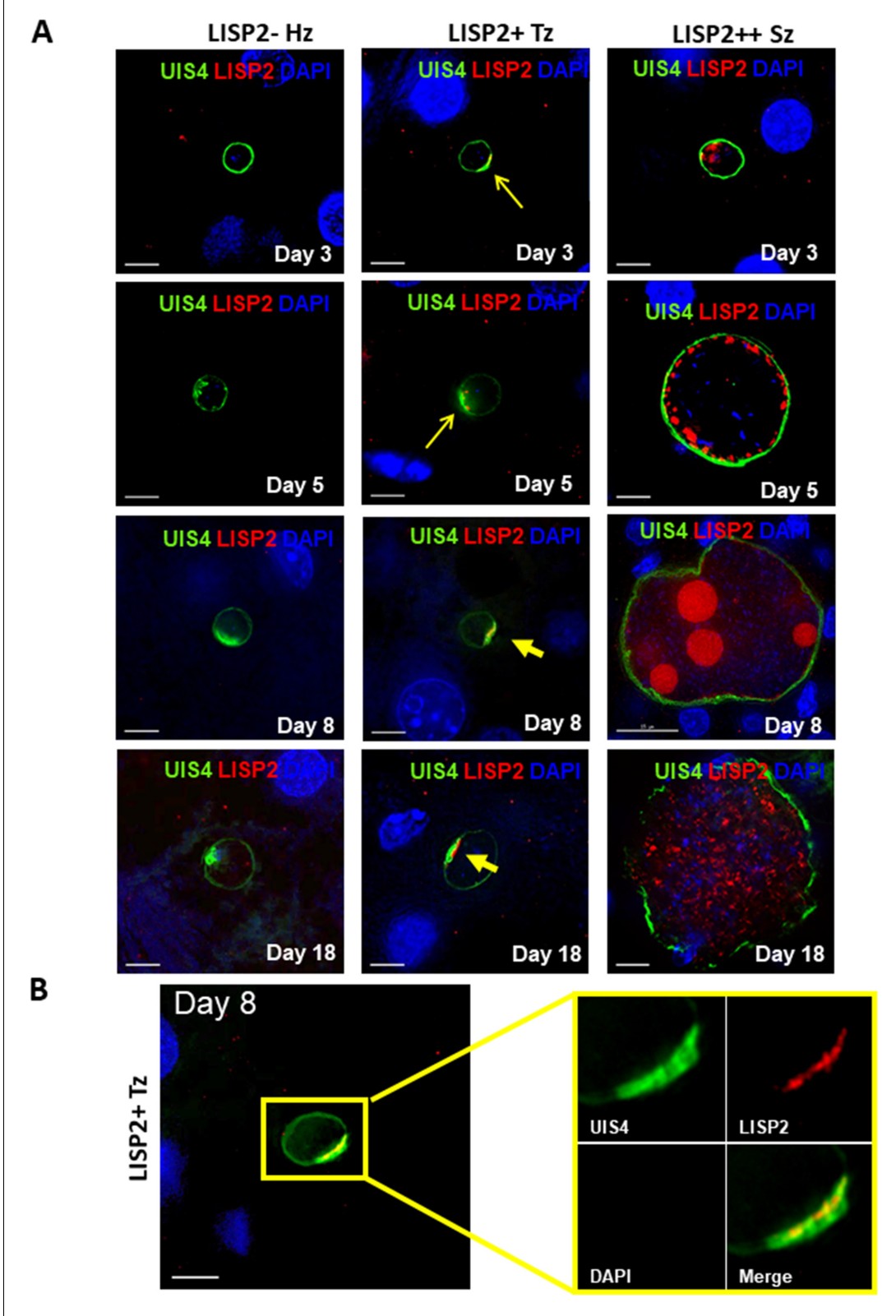

**Figure 2.** Evaluation of LISP2 expression in *P. vivax* infected huHep mice. (**A**) Hepatic stages of *P. vivax* infected huHep mice were stained with DAPI (blue), a monoclonal antibody specific for *P. vivax* LISP2 protein (red) and a monoclonal antibody specific for *P. vivax* UIS4 protein (green). Representative images of LISP2-, LISP2+ and LISP2++ parasites at days 3, 5, 8 and 18 post-sporozoite infection are shown. (**B**) Day 8 LISP2+ trophozoite

*Figure 2 continued on next page*

*Figure 2 continued*

visualized for LISP2 (red) and UIS4 (green). The area in the yellow box is highlighted to show partial overlap of LISP2 and UIS4 localization at or near PVM prominence. Scale bar, 25 µm.

DOI: https://doi.org/10.7554/eLife.43362.008

The following figure supplement is available for figure 2:

**Figure supplement 1.** UIS4 (up-regulated in infective sporozoites gene 4) protein localized at the PVM (Parasitophorous vacuole membrane) in *P. cynomolgi* liver stages.

DOI: https://doi.org/10.7554/eLife.43362.009

maximum proportion of 15–20% by day 4. From Day 5 onwards, the fraction of LISP2+ parasites remained small (≤5%) but steady. A small number of LISP2++ parasites (≤3%) was first detected around day 4 and the proportion of LISP2++ developing parasites increased to about 20% from days 5 to day 6 onwards (*Figure 3—figure supplement 1*).

We further demonstrated that these LISP2- parasites were viable parasites as they were labeled with the transcriptional activity marker H3K9ac (*Voorberg-van der Wel et al., 2017*; *Gupta et al., 2016*). Notwithstanding, as previously shown (*Voorberg-van der Wel et al., 2017*), schizonts were remarkably more transcriptionally active than hypnozoites as demonstrated by the much larger number of H3K9ac positive structures (*Figure 3—figure supplement 2* and *Video 1*).

Given the fact that LISP2 staining was first observed in characteristic crescent shape on PVM in LISP2+ parasites, we hypothesized that it could be a PV resident protein. To investigate the localization of LISP2 in PV, co-localization experiments were performed using antibodies developed against the *P. cynomolgi* ortholog of *P. falciparum* PV resident proteins; PV1 (parasitophorous vacuolar protein 1; PcyM_0933600). PV1 was previously reported to be resident protein of PV in blood stages (*Chu et al., 2011*). Further demonstrating that LISP2- stained parasites are not artifact staining but bona fide constructive infections, parasites were stained positively with PV1 and similarly to the staining pattern observed in blood stages PV1 antibodies specifically highlight the circumference of the parasitophorous vacuole (*Nyalwidhe and Lingelbach, 2006*) (*Figure 3c*). Interestingly, early in development the crescent of LISP2+ parasites overlapped with PV1 staining. Importantly, we observed that both LISP2- hypnozoites and LISP2+ trophozoites show identical PV1 staining patterns. Strikingly, PV1 protein co-localizes with LISP2 in LISP2++ schizonts, which suggests that LISP2 could be a PV resident protein (*Figure 3c*, *Figure 3—figure supplement 3* and *Videos 2*, *3*, *4* and *5*). Taken together, we show that LISP2- hypnozoites and the earliest developing form, LISP2+ trophozoites, are of similar size range, have a functional PV as they were stained positively for PV proteins, PV1 or UIS4 and could be distinguished on the basis of LISP2 expression.

## LISP2 expression correlates with drug sensitivity to prophylactic drugs

Based on the above results, we hypothesized that LISP2 expression may discriminate developing forms from persistent hypnozoites that do not react with LISP2 antibodies and could serve as a molecular marker to predict the anti-relapse drug effect in vitro assays. The most salient pharmacological features of dormant hypnozoites are: (1) susceptibility to 8-aminoquinolines compounds like tafenoquine (TQ) (*Campo et al., 2015*) and, (2) resistance to atovaquone (ATQ) (*Baggish and Hill, 2002*) and PI4K inhibitors (*Zeeman et al., 2014*; *Zeeman et al., 2016*). We thus set out to determine the susceptibility of liver stage parasites with (LISP2+, LISP2++) and without LISP2 (LISP2-) exposed to three drugs TQ, ATQ, and the *Plasmodium* PI4K inhibitor KDU691 (*McNamara et al., 2013*). For each three drugs, we first carried out drug assays at single concentration that we have previously shown to be effective in *P. cynomolgi* infected hepatocytes (*Zeeman et al., 2014*; *Zeeman et al., 2016*; *Dembele et al., 2011*). Importantly, all drugs were found to be non-cytotoxic to simian hepatocytes up to 10 µM (*Figure 4—figure supplement 1*). Parasites were exposed to drugs starting at day 4 through day 8 post-infection in a radical cure mode assay and parasite counts were carried out at day 8, day 15 and day 21 (*Figure 4A*).

At day 8, as expected, LISP2- hypnozoites were not sensitive to ATQ (0.25 µM) and KDU691 (0.5 µM) treatments (*Figure 4B*). In contrast, the 8-aminoquinoline, TQ (1 µM) was active against dormant LISP2- hypnozoites at all three time-points. (*Figure 4B and C*). In addition, compared to DMSO, TQ

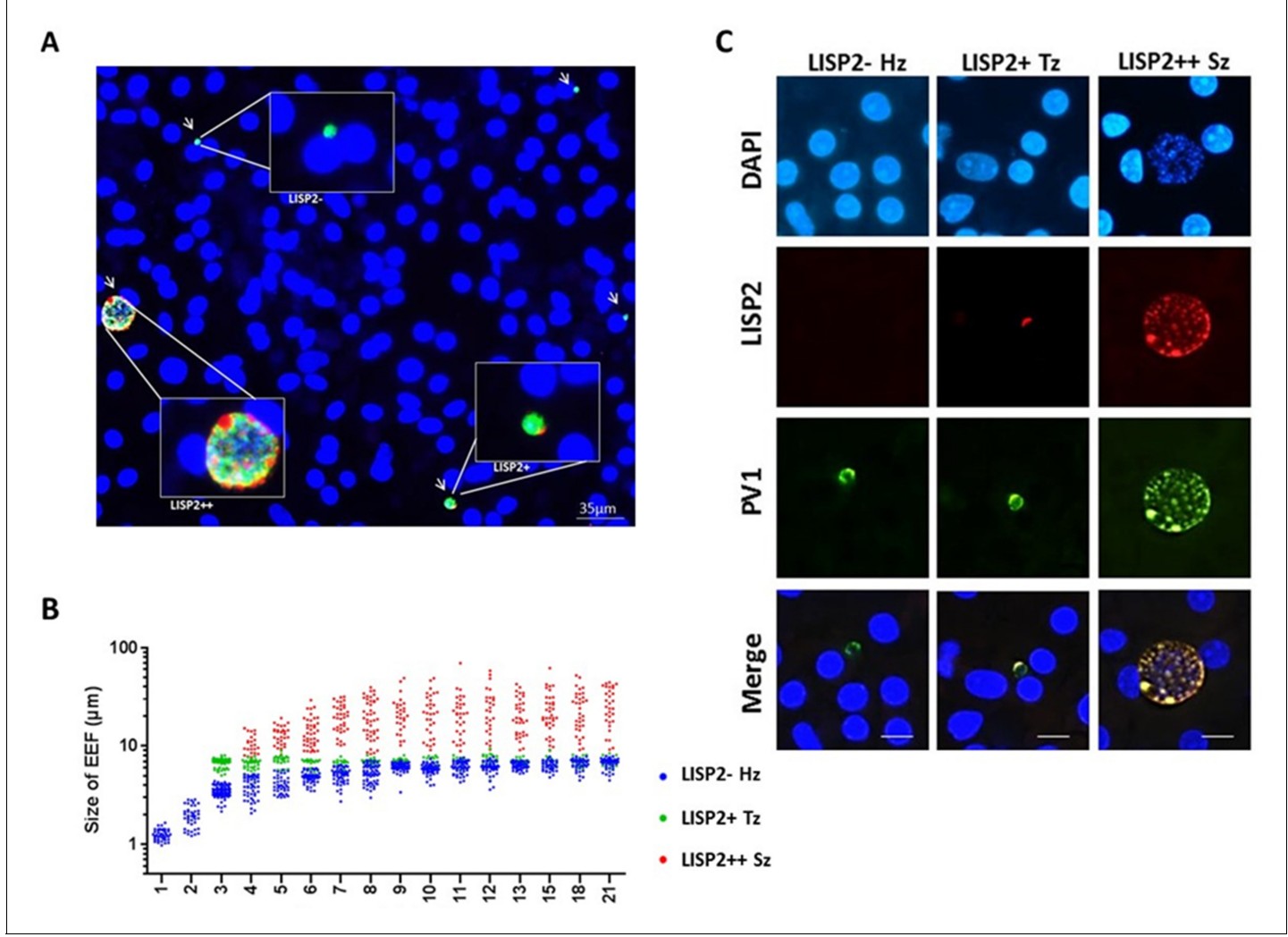

**Figure 3.** LISP2 expression distinguishes hypnozoites population from early developing liver stages. (**A**) Immunofluorescence assay (IFA) of *P. cynomolgi* liver stage parasites in simian primary hepatocytes 6 days after sporozoite inoculation visualized with DAPI for DNA content (blue), a polyclonal antibody specific for *P. cynomolgi* HSP70 protein (green), a monoclonal antibody specific for *P. cynomolgi* LISP2 protein (red). Scale bar, 35 µm. Merged IFA image showing *P. cynomolgi* three distinct hepatic parasite populations at day 6 post-infection: LISP2- (green), LISP2+ (weak red crescent staining) and LISP2++ (strong red peripheral and vacuolar staining). (**B**) Growth kinetic of three distinct *P. cynomolgi* parasite populations, dormant hypnozoites (LISP2-), early developing trophozoite (LISP2+) and growing schizonts (LISP2++). Data for size (the diameter of the parasite) are from three independent experiments. The cumulative number of cells measured in three biological experiments for each time point from day 1 to day 21 was ≥90. (**C**) *P. cynomolgi* hepatic stages stained with LISP2 and PV1 (parasitophorous vacuole 1), a marker of PV at day 6 post-sporozoite infection. LISP2 + trophozoite and LISP2++ schizonts showed co-localization of LISP2 and PV1. In contrast, hypnozoites were devoid of any signal from anti-Pc LISP2 and were stained only by anti-Pc PV1. Scale bar, 25 µm. The source data is available for *Figure 3B* (see source data file *Figure 3*).
DOI: https://doi.org/10.7554/eLife.43362.010

The following source data and figure supplements are available for figure 3:

**Source data 1.** Growth kinetics data of different liver populations for 21 days post-infection.
DOI: https://doi.org/10.7554/eLife.43362.015
**Figure supplement 1.** Relative quantitative analysis of LISP2 expressing parasites.
DOI: https://doi.org/10.7554/eLife.43362.011
**Figure supplement 1—source data 1.** Relative proportion of LISP2 expressing parasites and total number of parasites assayed for 21 days post-infection.
DOI: https://doi.org/10.7554/eLife.43362.012
**Figure supplement 2.** Assessment of viability of *P. cynomolgi* liver stages.
DOI: https://doi.org/10.7554/eLife.43362.013
**Figure supplement 3.** Co-localization of PV1 and LISP2 at parasitophorous vacuole.
*Figure 3 continued on next page*

*Figure 3 continued*

DOI: https://doi.org/10.7554/eLife.43362.014

reduced 70% (p value 0.0003), 78% (p value 0.0009) and 83% (p value 0.04) of all hepatic forms at day 8, day 15 and day 21, respectively.

ATQ and KDU691 eliminated 55% (p value is 0.01) and 84% (p value is 0.005) of LISP2 expressing hepatic stages at day 8. Importantly, by day 15 and day 21 ATQ and KDU691 treated cultures displayed a number of LISP2 expressing parasites comparable to untreated controls (DMSO) (*Figure 4C*). This, together with the steady decrease of the number of LISP2- hypnozoites, strongly suggests that persistent hypnozoites (LISP2-) re-activated between day 8 and day 21 to produce LISP2+ trophozoites which growth was unhindered after the drugs were washed away to develop into new replicating schizonts (LISP2++). In contrast, because TQ killed LISP2- hypnozoites at day 8, the drug effects of TQ was borne out at day 15 and day 21 with >90% significant reduction from control of TQ (p value for TQ at day 15 is 0.001), respectively (*Figure 4B*).

We confirmed the above observations in dose response experiments by assessing the number of parasites that survived in the culture exposed to eight different doses of ATQ, TQ and KDU691 from day 4 to day 8 with concentration ranging from 10 µM to 13 nM. The radical cure assay was performed in triplicates in three independent experiments. Our assay showed that when compared to LISP2 negative parasites, LISP2 expressing parasites were respectively 13 and 14 fold more susceptible to ATQ and KDU691 (*Figure 4D*). In contrast, TQ displayed similar activity against LISP2 expressing parasites and LISP2-negative parasites. Thus LISP2- parasites fulfilled the pharmacological criteria for persistent dormant hypnozoites, namely that they are susceptible to 8-aminoquinolines but phenotypically resistant to atovaquone and PI4K inhibitors.

## Discussion

The discovery of novel *P. vivax* radical cure drugs, defined as the prevention of relapse and complete elimination of all liver stages of the malaria parasite including the dormant persistent hypnozoites, remains a considerable challenge (*Campo et al., 2015*). Anti-malarial drug discovery efforts against established hypnozoites and thus preventing relapse is hampered by numerous roadblocks, including lack of high-throughput screening assays that can predict relapse. Most importantly, drug development is hindered by lack of tractable markers that can assess hypnozoite reactivation and monitor the therapeutic efficacy of anti-relapse compounds. The data presented here, and graphically summarized in *Figure 5* suggest that the LISP2 protein expression could be a powerful tool to distinguish dormant hypnozoites from activated hypnozoites that differentiated to developing forms in relapse drug assays.

Using LISP2 antibodies, we characterized the development of *P. cynomolgi* liver stages in long-term in vitro hepatic cultures. We show that LISP2 is first detected in a characteristic crescent-shape pattern in a subset of small parasites (LISP2+) at day 3 post sporozoite infection, suggesting that LISP2 expression marks the beginning of liver stage development. From day 4 to days 9–10, LISP2 expression increases as parasites mature and grow, and LISP2 distribution drastically expands throughout the parasite membrane and cytosolic vacuoles (LISP2++). The cellular function of these previously reported vacuoles (*Tarun et al., 2006*) is unclear but it has been proposed that this is the cellular compartment in which LISP2 6-cys domain is cleaved and mediates LISP2 N-terminal region export to the hepatocyte cytosol through its signal peptide (*Orito et al., 2013*). Throughout the 21-day period, we found that a significant proportion of

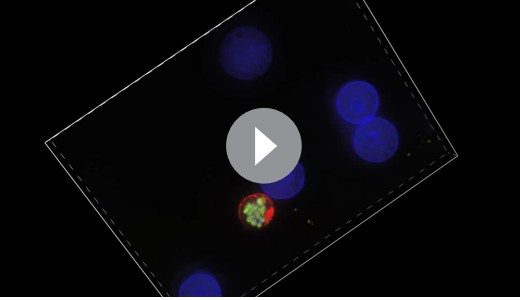

**Video 1.** Co-localization of H3K9ac and LISP2. Reconstructed 3-D movie of schizonts using de-convoluted Z stack images co-stained with DAPI (blue color), H3K9ac (green color) and LISP2 (red color).
DOI: https://doi.org/10.7554/eLife.43362.016

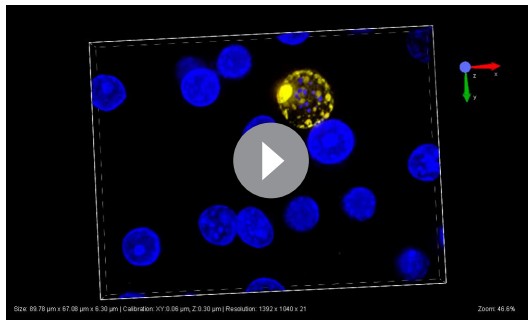

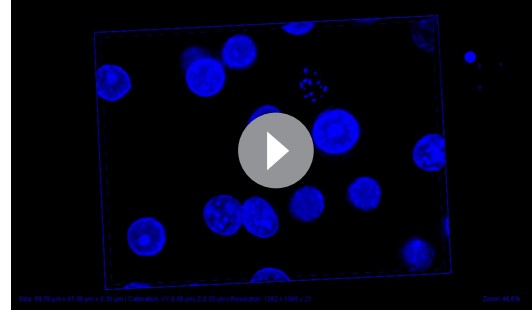

**Video 2.** Co-localization of PV1 and LISP2 at parasitophorous vacuole. Day 6 infected hepatocytes co-stained with DAPI (blue; see *Video 3*), PV1 (green; see *Video 4*) and LISP2 (red; see *Video 5*). 30–40 stacks of images were acquired with 100X objective lens with Z-step size of 0.3 µm. Three-Dimensional movies of the merged channel (in yellow color) and individual channels reconstructed using deconvoluted z-stack images are shown.

DOI: https://doi.org/10.7554/eLife.43362.017

**Video 3.** Three-dimensional movie of the day 6 infected hepatocytes stained with DAPI (blue). Each movie is made from 30 to 40 stacks of images acquired with 100X objective lens with Z-step size of 0.3 µm.

DOI: https://doi.org/10.7554/eLife.43362.018

persistent small parasites do not express LISP2 (LISP2-). This LISP2 pattern of developmental expression in liver-stages is fully recapitulated in vivo in the liver of *P. vivax* infected human-chimeric liver mice. We also observed that in early developing parasites (LISP2+ trophozoites), UIS4 and LISP2 co-localize at the previously reported *P. vivax* parasitophorous vacuole membrane (PVM) prominence which was proposed to play a crucial role in the biogenesis and initial development of liver stages (*Mikolajczak et al., 2015*). Importantly, LISP2- hypnozoites and LISP2+ trophozoites appear as two different subpopulations of parasites which are of similar size range and both exhibit the UIS4-prominence characteristic of hypnozoites (*Mikolajczak et al., 2015*). Although the UIS4 prominence was initially regarded as a marker of established *P. vivax* hypnozoites, recent reports have shown that it is also present in small developing forms (*Gural et al., 2018*). We could confirm these observations in *P. cynomolgi* parasites and further establish that the presence of LISP2 in the UIS4 prominence uniquely distinguishes the early trophozoites from the similarly sized population of established hypnozoites that remain LISP2 negative.

In *P. cynomolgi* liver stages we found that the PV protein PV1 (*Morita et al., 2018*), localizes to the parasitophorous vacuole, which is a critical interface that separates parasite from host hepatocyte cytoplasm (*Mueller et al., 2005*). Interestingly, we found that LISP2 and PV1 proteins extensively co-localize throughout schizogony, from early developing trophozoites (LISP2+) to late schizonts (LISP2++). PV1 interacts with the export machinery required for the trafficking of *Plasmodium* proteins across PVM to infected erythrocytes during blood stages (*Morita et al., 2018*) and further functional studies will be required to determine whether LISP2 and PV1 play similar roles in liver stages.

In both, in vitro P. cynomolgi culture and in vivo P. vivax model, we observe LISP2+ trophozoites throughout the time course of our studies. Although we cannot entirely rule out that some of the LISP2+ trophozoites observed are developmentally arrested forms, our data showing (1), active transcription with positive H3K9 staining and (2), the emergence of LISP2+ trophozoites after prophylactic drug treatments, strongly suggest that LISP2+ trophozoites result from hypnozoite activation or relapse. Given the

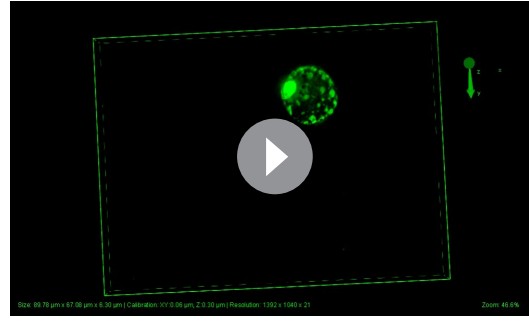

**Video 4.** Three-dimensional movie of the day 6 infected hepatocytes stained with PV1 (green). Each movie is made from 30 to 40 stacks of images acquired with 100X objective lens with Z-step size of 0.3 µm.

DOI: https://doi.org/10.7554/eLife.43362.019

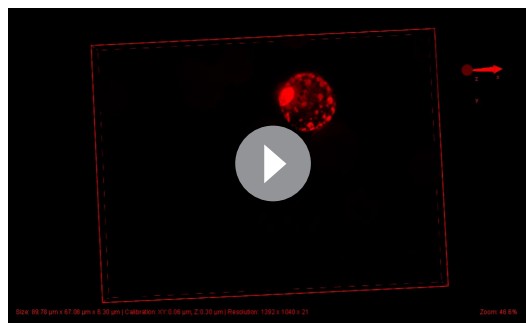

**Video 5.** Three-dimensional movie of the day 6 infected hepatocytes stained with LISP2 (red). Each movie is made from 30 to 40 stacks of images acquired with 100X objective lens with Z-step size of 0.3 μm.
DOI: https://doi.org/10.7554/eLife.43362.020

large difference in the inoculum used in our experimental models it is however not clear how the observations reported above relate to clinical malaria relapse rates. Indeed, it is well known that the size of the sporozoites inoculum critically influences the relapse rate observed in simian *P. cynomolgi* malaria (*Schmidt, 1986*).

One limitation of our study is that we could not monitor LISP2 expression in real-time at the cellular level to unambiguously establish the progressive transition from LISP2- persistent hypnozoites, to LISP2+ trophozoites. Nonetheless, the kinetics of LISP2 expression in vitro (*P. cynomolgi*) and in vivo (*P. vivax*) suggests that LISP2 protein expression is an early event of liver-stage development. Genetic engineering of GFP-expressing parasites reporting the protein localization of LISP2 over time will more firmly establish this developmental sequence and it will help to better delineate the transition from activating hypnozoite to early schizont.

Finally, we show that in *P. cynomolgi* infected hepatocytes, the radical cure drug TQ was potently active on all hepatic stages irrespective of LISP2 expression. In sharp contrast KDU691, a selective inhibitor of the PI4K enzyme, potently clears LISP2 positive (LISP2+ and LISP2++) parasites, but is inactive on persistent LISP2-negative (LISP2-) parasites.

Taken together our data show that the LISP2- parasites display the morphological characteristics and the pharmacological susceptibility profile—sensitivity to 8-aminoquinoline drugs and phenotypic resistance to the prophylactic drug KDU691—of dormant hypnozoites. LISP2 expression closely coincides with the beginning of liver-stage development and is a marker of malaria relapse. These findings will enable the design of novel in vitro assays that directly measure the anti-relapse drug effects in vitro by measuring and following LISP2 activation. Finally, and importantly, our study provides an additional molecular feature to our definition of *Plasmodium* hypnozoites, now defined as small, uninuclear persistent hepatic parasites with a UIS4-prominence lacking the LISP2 protein.

# Materials and methods

## Key resources table

| Reagent type (species) or resource | Designation | Source or reference | Identifiers | Additional information |
|---|---|---|---|---|
| Biological sample (*Macaca mulatta*) | Primary simian hepatocytes | Biomedical Primate Research Centre, Netherlands | | Freshly isolated from Macaca mulatta, |
| Biological sample (*Macaca fascicularis*) | Primary simian hepatocytes | SingHealth, Singapore | | Freshly isolated from Macaca fascicularis |
| Biological sample (*P. cynomolgi*) | sporozoites | Armed forces research institute of medical sciences (AFRIMS) Bangkok | | *Anopheles dirus* mosquito infected with *P. cynomolgi* |
| Biological sample (*P. cynomolgi*) | sporozoites | Biomedical Primate Research Centre, Netherlands | | *Anopheles stephensi* mosquito infected with *P. cynomolgi* |
| Biological sample (*P. vivax*) | sporozoites | Mahidol University | | *Anopheles dirus* mosquito infected with *P. vivax* |

*Continued on next page*

*Continued*

| Reagent type (species) or resource | Designation | Source or reference | Identifiers | Additional information |
|---|---|---|---|---|
| Antibody | Mouse anti-LISP2 antibody (monoclonal) | this paper | | synthesized by genscript; IFA (1:500) |
| Antibody | Rabbit anti-PV1 antibody (polyclonal) | this paper | | synthesized by genscript; IFA (1:500) |
| Antibody | Rabbit anti-HSP70 antibody (polyclonal) | this paper | | synthesized by genscript; IFA (1:2000) |
| Antibody | Rabbit anti-H3K9ac antibody (monoclonal) | Abcam, | ab177177 | IFA (1:1000) |
| Antibody | Alexa 488-conjugated goat anti-rabbit immunoglobulin (polyclonal) | Invitrogen | 11034 | IFA (1:4000) |
| Antibody | Alexa 594-conjugated goat anti-mouse immunoglobulin (polyclonal) | Invitrogen | 11032 | IFA (1:5000) |
| Sequence-based reagent | RNA FISH probes | this paper | | Synthesized by Advanced Cell Diagnostics |
| Commercial assay or kit | CCK-8 kit | Sigma | 96992 100TESTS-F | |
| Chemical compound, drug | KDU691 | Novartis internal chemical library | | Synthesized by Novartis |
| Chemical compound, drug | Atovaquone (ATQ) | Novartis internal chemical library | | Synthesized by Novartis |
| Chemical compound, drug | Tafenoquine (TQ) | Novartis internal chemical library | | Synthesized by Novartis |
| Chemical compound, drug | DMSO | Sigma | D8418 | |

## Ethics statement

Nonhuman primates were used because no other models (in vitro or in vivo) were suitable for the aims of this project. NHP were used at the Biomedical Primate Research Centre (BPRC), The Netherlands, Novartis Laboratory of Large Animal Services, New Jersey, U.S.A., (Novartis-LAS), SingHealth, Singapore, and The Armed Forces Research Institute of Medical Sciences, Bangkok, Thailand (AFRIMS).

The local independent ethical committee constituted conform Dutch law (BPRC Dier Experimenten Commissie, DEC) approved the research protocol (agreement number DEC# 750) prior to the start and the experiments were all performed according to Dutch and European laws. The Council of the Association for Assessment and Accreditation of Laboratory Animal Care (AAALAC International) has awarded BPRC full accreditation. Thus, BPRC is fully compliant with the international demands on animal studies and welfare as set forth by the European Council Directive 2010/63/EU, and Convention ETS 123, including the revised Appendix A as well as the 'Standard for humane care and use of Laboratory Animals by Foreign institutions' identification number A5539-01, provided by the Department of Health and Human Services of the United States of America's National Institutes of Health (NIH) and Dutch implementing legislation. The rhesus monkeys (*Macaca mulatta*, either gender, age 4–7 years, Indian or mixed origin) used in this study were captive-bred and socially housed. Animal housing was according to international guidelines for nonhuman primate care and use. Besides their standard feeding regime, and drinking water ad libitum via an automatic watering

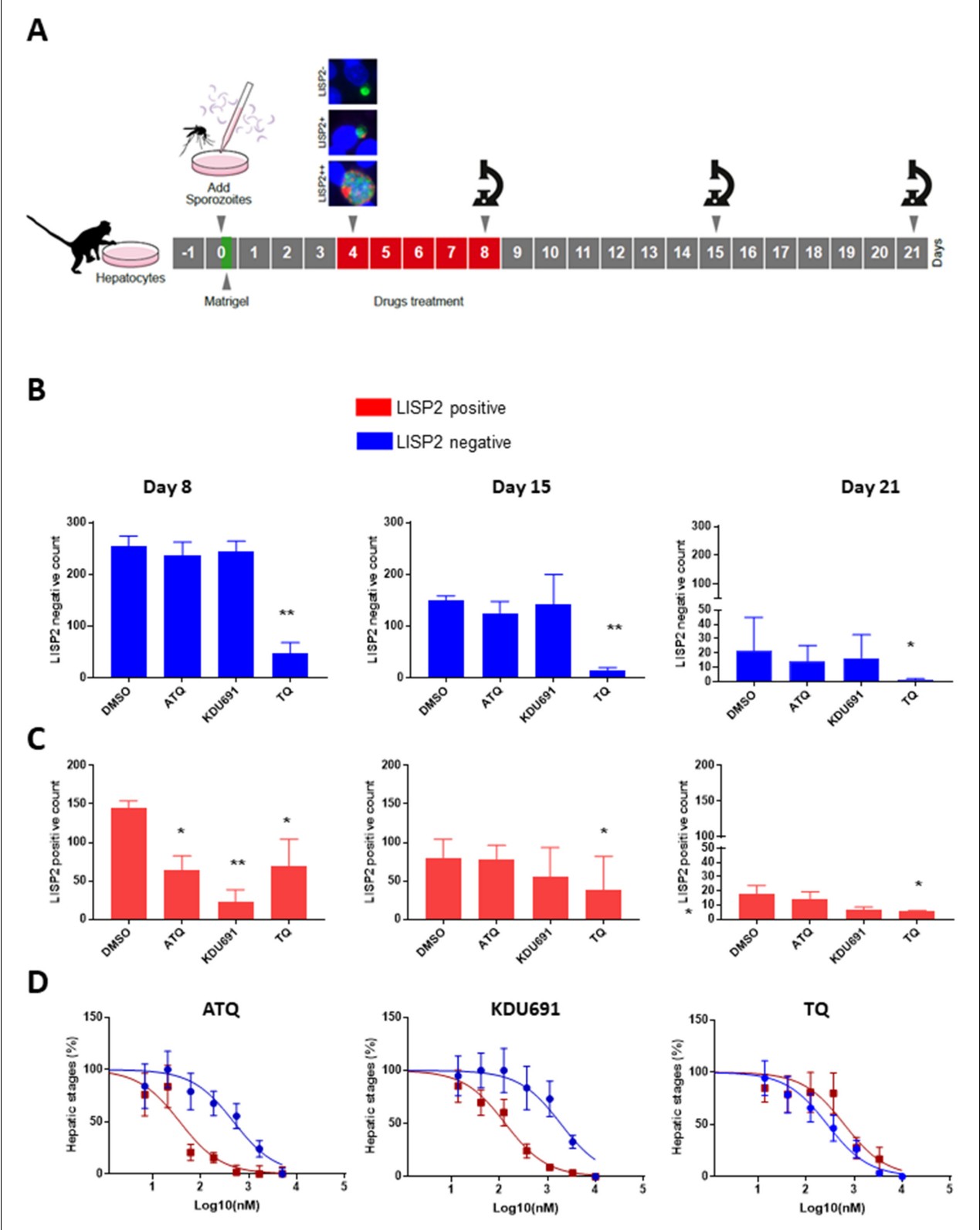

**Figure 4.** LISP2 negative parasites are refractory to PI4K inhibition but sensitive to 8-aminoquinoline drugs. (A) Simian primary hepatocytes infected with *P. cynomolgi* parasites were exposed to ATQ (0.25 µM), KDU691 (0.5 µM) and TQ (1 µM) from day 4 to day 8 post-infection. The effects of drug treatments were compared to untreated control (DMSO) at day 8, day 15 and day 21 post-infection; p value is assessed by statistical one–tailed t test with *p<0.05 and**0.01, respectively. (B) Blue bars show drug effects on persistent LISP2- hypnozoites. (C) Red bars represent drug effects on LISP2-

*Figure 4 continued on next page*

Figure 4 continued

positive parasites. (D) Dose response curve for ATQ, KDU691 and TQ are shown. The blue curves represent drug activity against hypnozoite (LISP2-) and red curves show activity against LISP2 positive parasites. The IC50 (inhibitory concentration) of KDU691 against LISP2- hypnozoites and LISP2 expressing parasites were 1.8 µM and 0.13 µM. The IC50 of ATQ for LISP2- and LISP2 expressing parasites were 0.48 µM and 0.036 µM, respectively. The graph bar in **Figure 4B and C** represent mean with standard deviation (s.d) from five technical replicates in two independent biological assays, whereas in **Figure 4D** from three independent biological assays. The source data is available for **Figure 4** (see source data file **Figure 4**).
DOI: https://doi.org/10.7554/eLife.43362.021
The following source data and figure supplements are available for figure 4:

**Source data 1.** In vitro radical cure mode assay for liver stage.
DOI: https://doi.org/10.7554/eLife.43362.024
**Figure supplement 1.** Cytotoxic cell survival assay.
DOI: https://doi.org/10.7554/eLife.43362.022
**Figure supplement 1—source data 1.** Cell cytotoxicity assay data.
DOI: https://doi.org/10.7554/eLife.43362.023

system, the animals followed an environmental enrichment program in which, next to permanent and rotating non-food enrichment, an item of food-enrichment was offered to the macaques daily. All animals were monitored daily for health and discomfort. All intravenous injections and large blood collections were performed under ketamine sedation, and all efforts were made to minimize suffering. Liver lobes were collected from monkeys that were euthanized in the course of unrelated studies (ethically approved by the BPRC DEC) or euthanized for medical reasons, as assessed by a veterinarian. Therefore, none of the animals from which liver lobes were derived were specifically used for this work, according to the 3Rrule thereby reducing the numbers of animals used. Euthanasia was performed under ketamine sedation (10 mg/kg) and was induced by intracardiac injection of euthasol 20%, containing pentobarbital.

*Macaca fascicularis* (cynomolgous monkeys) were used at the other facilities for the same purpose. Studies were approved by the SingHealth Institutional Animal Care And Use Committee (IACUC), Ref No.: 2015/SHS/1024, Novartis IACUC, protocol number 100280 and AFRIMS IACUC, Protocol Number PN13-06. AFRIMS, SingHealth and Novartis-LAS have full AAALAC accreditation.

### *Plasmodium cynomolgi* sporozoite production and sporozoite isolation

*Macaca fascicularis* was infected with *P. cynomolgi bastianellii* (also known as strain B) sporozoites and blood stage parasitemia monitored by Giemsa stained smears. *Anopheles dirus* mosquito were directly fed on blood meal from the infected monkey. Salivary glands were collected 14–30 days post-mosquito infection. Infected mosquitoes salivary glands removal and sporozoite recovery was done as previously reported (*Dembélé et al., 2014*). Sporozoites of *P. cynomolgi* were isolated from infected *A. dirus* salivary glands removed by hand dissection. Salivary glands were crushed in a potter; sporozoite form parasites were recovered after filtration through a 40 µm filter (Cell Strainer, Becton Dickinson) and a 3-min centrifugation at 4°C; $12 \times 10^3$ rpm. *P. cynomolgi* infected *A. dirus* mosquitoes were obtained from the United States army medical directorate; armed forces research institute of medical sciences (AFRIMS) Bangkok 10400, Thailand. We have used *P. cynomolgi* M strain and *A. stephensi* for RNA-FISH experiments. Generation of *P. vivax* sporozoites, infection of huHep mice with *P. vivax* sporozoites and isolation of liver stages were carried out as described earlier (*Mikolajczak et al., 2015*).

### Simian primary hepatocyte culture conditions and sporozoite infection

The primary hepatocytes used in this study were freshly isolated from healthy *Macaca fascicularis* liver segments as previously described (*Silvie et al., 2003*). For each experiment, at day −1, frozen hepatocytes were first thawed directly in 37°C water bath and then transferred into 25 mL of a pre-warmed recovery Medium A (Life technologies, CM7000). Hepatocytes were spun down at 1500 rpm for 5 min. Media were removed and the cell pellet re-suspended into 8 mL of Medium B (hepatocytes plating medium containing William E without phenol red indicator, Life Technologies ref: A1217601, supplemented with Life technologies supplement Pack CM3000, Life technologies CM3000). Hepatocytes were seeded at a density $25 \times 10^4$ cells per cm (*Armistead and Adams, 2018*) in collagen I coated plates 96 wells (Greiner bio One ref 655956) and 48-well (Nest Scientific

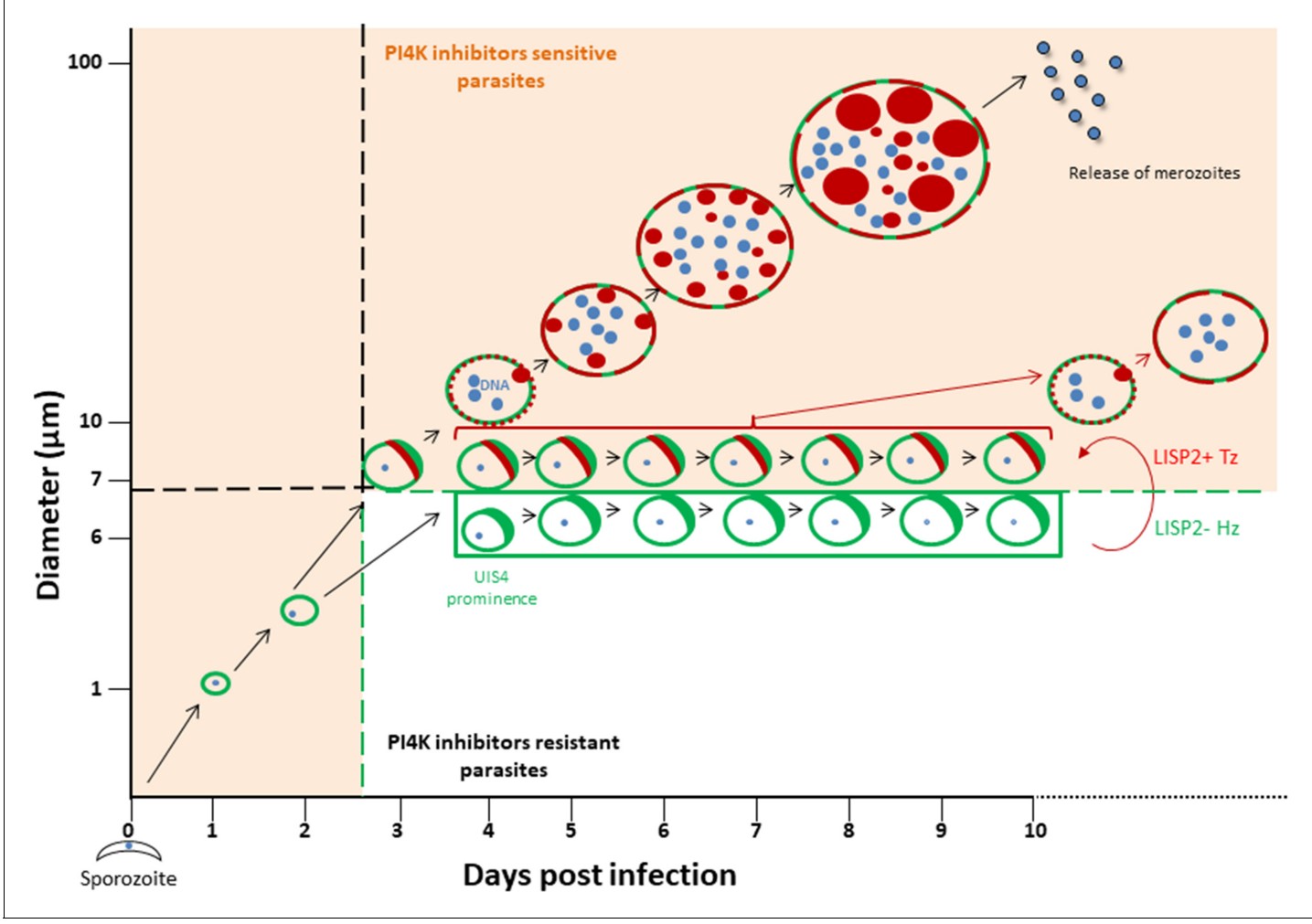

**Figure 5.** Schematic model to distinguish hypnozoites from other liver stages based on PI4K drug susceptibility. Malaria liver stage infection begins when infective sporozoites injected by mosquitoes invade host hepatocytes. Within hepatocytes, sporozoites transform into hepatic schizonts for 7–9 days and subsequently, mature schizonts burst and release pathogenic merozoites into blood circulation (**Prudêncio et al., 2011**). Sporozoite transformation into early liver stage development coincides with LISP2 expression. LISP2 expression marks the beginning of liver stage development at day 3 and its expression increases as liver stage progresses into complete maturation. Sporozoites of relapsing malaria species such as *P. vivax* and *P. cynomolgi* can generate dormant hypnozoites (**Wells et al., 2010**). LISP2- Hz exhibits UIS4 prominence on PVM (shown in green) and grows till day 5 and remains persistent and unchanged from day 5 onwards. LISP2 +Tz display crescent staining pattern (shown in red) which colocalizes with UIS4 protein on PVM. Beyond day 6 post-sporozoite infection, both LISP2- Hz and LISP2+ Tz have identical size and can only be discriminated on the basis of susceptibility to PI4K inhibitor. The shaded areas (in orange) represent the parasite that can be eliminated by both PI4K inhibitors and 8-aminoquinoline drugs. The non-shaded area in the model represents only LISP2- Hz which is resistant to PI4K and can be eliminated only by 8-aminoquinoline drugs. List of supplemental tables.
DOI: https://doi.org/10.7554/eLife.43362.025

748001); then incubated in 37°C, 5% $CO_2$ for overnight. At day-0, Medium C (William's E medium with phenol red indicator, Life Technologies ref: 12551–032, supplemented with 10% FetalCloneII serum, ref: SH30066.03, $5 \times 10^{-5}$ M water-soluble hydrocortisone, sigma ref: H0396, 5 μg per ml insulin, 2 mM L-glutamine and 200 U per ml penicillin, 200 μg per ml streptomycin, Life Technologies) was used to wash away unattached cells and to maintain the culture. At day-0, *P. cynomolgi* freshly isolated sporozoites were re-suspended in the Medium C at $10^6$ sporozoites per mL. $6 \times 10^4$ and $12 \times 10^4$ sporozoites per well were inoculated into hepatocytes cultures in 96- and 48-well plates, respectively. To sediment parasite on the cells, infected culture plates were centrifuged for 10 min at 8°C, $2 \times 10^3$ rpm. Cultures were then incubated at 37°C, 5% $CO_2$ for 3 hr. At 3 hr post-infection, cultures were washed three times with Medium C and then maintained at 37°C, 5% $CO_2$ in

the same Medium C. For sandwich Matrigel cultures, Matrigel (BD Biosciences ref: 356234) was added as per the manufacturer's recommendations at day-0, 3 hr post-infection. To assess hepatic parasites growth kinetics and dynamics from day-0 to day-21, *P. cynomolgi* infected cultures were maintained in Medium C that was renewed every other day. For each defined time points, the cultures were ended by 100% cold methanol fixation for 3 min.

## Drug evaluation against *P. cynomolgi* hepatic stages using LISP2

PI4K inhibitor, KDU691 was synthesized as previously reported (*McNamara et al., 2013*) and all other compounds used in this study were procured from Sigma-Aldrich (St. Louis, MO). *P. cynomolgi* hepatic forms drug assessment was done as schematically shown in *Figure 4a* by treating culture from day 4 to day 8 with drugs renewed at day 6. At day 8, all drugs were removed by three consecutives washings and growth monitored for defined time points in medium C which was renewed every other day. Reference antimalarials which included non anti-relapsing compounds such as atovaquone (ATQ) (0.25 μM) and KDU691 (0.5 μM); the 8-aminoquinolines tafenoquine (TQ) (1 μM) and primaquine (PQ) (5 μM) that prevent relapse were used. These concentrations used were chosen for ATQ and KDU691 to selectively kill developing parasites (LISP2+ and LISP2++) and assess hypnozoite (LISP-) activation at the later days. For TQ and PQ, selected concentrations were to inhibit all liver stage parasites preventing later LISP2- parasites activation. All concentrations were selected based on compound dose response curves. Compounds $IC_{50}$ were generated by treating cultures as above with three-fold dilution concentrations from 10 μM to 13 nM. For ATQ, the three-fold dilution concentration was from 5 μM to 6 nM. For $IC_{50}$ generation, all cultures were ended at day 8 postinfection. One tail t test was used to assess the level of significant reduction of different hepatic forms due to drug treatment compared to DMSO treatment.

## Assessment of cellular viability of simian primary hepatocytes under drug exposure

To evaluate compounds cytotoxicity against the primary hepatocytes, the cells were cultured without infection and exposed to 10 μM of each compound treatment as above from day 4 to day 8 with all drugs renewed at day 6. Puromycin (Pur) was included as positive control (*Azzam and Algranati, 1973*) and DMSO as negative control during the drug exposure. To stop all cell metabolisms, five minutes of 100% methanol treatment was included at day 8. At day 8, cell counting kit-8 solution (CCK-8) was diluted 2.5-fold in complete Medium C and 70 μL of diluted CCK8 was added to each well of the plate containing 50 μL of Medium C. The plates were incubated for 3 hr at 37°C, 5% $CO_2$. The absorbance at 450 nm was measured by Envision.

## Parasite immuno-fluorescence assay (IFA) and RNA FISH

*P. cynomolgi* liver stages were detected with anti-*P. cynomolgi* HSP70 (PcyM_0515400) polyclonal antibody, anti-*P. cynomolgi* LISP2 (PcyM_0307500) monoclonal antibody, anti-H3K9ac monoclonal antibody (Abcam, ab177177) and anti-*P. cynomolgi* PV1 (PcyM_0933600) polyclonal antibody. The polyclonal and monoclonal antibodies used in our study were synthesized by Genscript. Briefly, using codon optimization and BacPowerTM bacterial protein expression technology, the selected epitopes were synthesized and used for immunization of animals as per Genscript established methods. Following 100% cold methanol fixation at the defined time points, *P. cynomolgi* cultures were first incubated for two hours at 37°C with the primary antibodies. Anti-*P. cynomolgi* HSP70 polyclonal antibody diluted 1:2000 in 1XPBS, anti-*P. cynomolgi* LISP2 monoclonal antibody diluted 1: 500 in 1XPBS, anti-H3K9ac monoclonal antibody diluted in 1:1000 in 1X PBS and anti-*P. cynomolgi* PV1 polyclonal antibody diluted 1: 500 in 1XPBS. Subsequently to 2 hr of incubation with these antibodies was followed by three washing with 1X PBS. To reveal hepatic stages parasites, cultures were then incubated with fluorescent secondary antibodies Alexa 488-conjugated goat anti-rabbit immunoglobulin (11034 Invitrogen) and Alexa 594-conjugated goat anti-mouse immunoglobulin and supplemented with 1 μg per ml of DAPI (Sigma) to stain parasites and cells nuclei for 1 hr. Secondary antibodies were removed by three washing with 1X PBS. Infected plates were kept with 1X PBS at 4°C for further parasite imaging and quantification. RNA-FISH was performed in the 96-well plates using the RNAscope Multiplex Fluorescent Assay v2 kit from Advanced Cell Diagnostics essentially according to the manufacturer's instructions. Following rehydration protease digestions were

performed using pretreatment solution three from the kit at 1:10 dilution for 20 min at room temperature. Hybridizations were 2 hr at 40°C. Probes used were directed against *P. cynomolgi* HSP70 (PcyM_0515400, region 606–1837 of XM_004221103.1) and LISP2 (PcyM_0307500, region 2866–3853 of XM_004220801.1). After hybridization, TSA amplification steps were performed as described by the manufacturer. Following DAPI staining, cells were kept in PBS for imaging or immunofluorescence analysis. Images were acquired with a Leica DMI6000B inverted fluorescence microscope equipped with a DFC365FX camera using a HC PL APO 63x/1.40–0.60 oil objective. To evaluate specificity of RNA FISH signal, we have carried out RNAse A treatment before hybridization and we observe total loss of signal as compared to without RNAse A treatment (see *Figure 1—figure supplement 5*). This negative control confirms the specificity of the LISP2 signal.

For visualization of *P. vivax* liver stages in huHep mouse livers, livers were harvested and fixed for 24 hr in 4% paraformaldehyde. Liver lobes were cut into 50 μm sections using a Vibratome apparatus (Ted Pella Inc, Redding, CA). For IFA, sections were permeabilized in Tris buffered saline (TBS) containing 3% $H_2O_2$ and 0.25% Triton X-100 for 30 min at room temperature. Sections were then blocked in TBS containing 5% dried milk at least 1 hr and incubated with primary antibody at 4°C overnight. *P. vivax* liver stages were detected with polyclonal LISP2 antibody (PVX_000975) (1:200) and monoclonal antibody specific for *P. vivax* UIS4 protein (PVX_001715) (1:500). Secondary antibodies were incubated for 2 hr at room temperature. Fluorescence images were acquired using an Olympus Delta Vision microscope equipped with deconvolution software.

### Parasite imaging and quantification
Enumeration of *P. cynomolgi* liver stage parasites was done under a fluorescence NIKON ECLIPSE TS100 microscope with a 40X magnification. Images were obtained from LEICA DM4000B. GraphPad Prism7 and Microsoft excel were used to generate all graph figure and dose response curves. ANOVA Statistical test was used to compare the compound toxicity to reference toxic compound Puromycin. The populations compared were from three independent biological replicates. A p value of 0.05 or less was considered to be statistically significant. Hepatic stages were measured in diameter (μm) as reported earlier (*March et al., 2013*; *Zeeman et al., 2014*).

## Acknowledgements
We thank Rochanawan Sootichote of the United States army medical directorate; armed forces research institute of medical sciences (AFRIMS) Bangkok 10400, Thailand for providing *A. dirus* mosquitoes. We thank the members of the malaria team from Novartis Institute for Tropical diseases for skillful technical assistance for mosquito dissection. We thank Takafumi Tsuboi of Ehime University, Japan for technical support for *P. vivax* infection. We also thank Stephanie Moquin of Novartis Institute for Biomedical Research for her support during initial set up for performing western blot experiments. This work has been funded by the Bill and Melinda Gates foundation (OPP1141292 and OPP1137694).

## Additional information

### Competing interests
Devendra Kumar Gupta, Guglielmo Roma, Andy Yip, Erika L Flannery, Sebastian Mikolajczak, Thierry Tidiane Diagana: is employed by and/or is a shareholder of Novartis Pharma AG. Pablo Bifani: employed by and/or shareholder of Novartis Pharma AG. The author declares no other competing interests exist. The other authors declare that no competing interests exist.

### Funding

| Funder | Grant reference number | Author |
|---|---|---|
| Bill and Melinda Gates Foundation | OPP1141292 | Guglielmo Roma<br>Clemens HM Kocken<br>Thierry Tidiane Diagana |

| Bill and Melinda Gates Foundation | OPP1137694 | Sebastian Mikolajczak |
| --- | --- | --- |

The funders had no role in study design, data collection and interpretation, or the decision to submit the work for publication.

## Author contributions

Devendra Kumar Gupta, Thierry Tidiane Diagana, Conceptualization, Resources, Data curation, Software, Formal analysis, Supervision, Validation, Investigation, Visualization, Methodology, Writing—original draft, Project administration, Writing—review and editing; Laurent Dembele, Resources, Data curation, Software, Formal analysis, Supervision, Visualization, Validation, Writing—original draft, Project administration, Writing—review and editing; Annemarie Voorberg-van der Wel, Guglielmo Roma, Vorada Chuenchob, Ashley M Vaughan, Stefan H Kappe, Validation, Investigation, Methodology, Writing—review and editing; Andy Yip, Niwat Kangwanrangsan, Erika L Flannery, Jetsumon Sattabongkot, Tomoko Ishino, Investigation, Methodology; Sebastian Mikolajczak, Pablo Bifani, Clemens HM Kocken, Resources, Supervision, Investigation, Methodology, Project administration, Writing—review and editing, Funding acquisition

## Author ORCIDs

Devendra Kumar Gupta (ID) https://orcid.org/0000-0001-6594-3980
Annemarie Voorberg-van der Wel (ID) https://orcid.org/0000-0001-9403-0515
Guglielmo Roma (ID) http://orcid.org/0000-0002-8020-4219
Tomoko Ishino (ID) http://orcid.org/0000-0003-2466-711X
Stefan H Kappe (ID) http://orcid.org/0000-0003-1540-1731
Erika L Flannery (ID) https://orcid.org/0000-0003-0665-7954
Jetsumon Sattabongkot (ID) http://orcid.org/0000-0002-3938-4588
Thierry Tidiane Diagana (ID) https://orcid.org/0000-0002-8520-5683

## Ethics

Animal experimentation: Nonhuman primates were used because no other models (in vitro or in vivo) were suitable for the aims of this project. NHP were used at the Biomedical Primate Research Centre (BPRC), The Netherlands, Novartis Laboratory of Large Animal Services, New Jersey, U.S.A., (Novartis-LAS), SingHealth, Singapore, and The Armed Forces Research Institute of Medical Sciences, Bangkok, Thailand (AFRIMS). The local independent ethical committee constituted conform Dutch law (BPRC Dier Experimenten Commissie, DEC) approved the research protocol (agreement number DEC# 750) prior to the start and the experiments were all performed according to Dutch and European laws. Further details are included in the Methods section of the paper.

## Decision letter and Author response

Decision letter https://doi.org/10.7554/eLife.43362.032
Author response https://doi.org/10.7554/eLife.43362.033

# Additional files

## Supplementary files

• Supplementary file 1. List of genes that were detected above >10 FPKM (Fragments Per Kilobase per Million) in hypnozoite (727 genes) and that were in the top 5% genes expressed in schizonts (134 genes). The data listed in the table derived from Voorberg et al, 2017. (Sz is schizont; three replicates and Hz is hypnozoite; four replicates)
DOI: https://doi.org/10.7554/eLife.43362.026

• Transparent reporting form
DOI: https://doi.org/10.7554/eLife.43362.027

## Data availability

All data generated during the study are submitted as supplementary source files.

The following previously published dataset was used:

| Author(s) | Year | Dataset title | Dataset URL | Database and Identifier |
|-----------|------|---------------|-------------|-------------------------|
| Annemarie Voorberg-van der Wel, Guglielmo Roma, Devendra Kumar Gupta, Sven Schuierer, Florian Nigsch, Walter Carbone, Anne-Marie Zeeman, Boon Heng Lee, Sam O. Hofman, Bart W. Faber, Judith Knehr, Erica M. Pasini, Bernd Kinzel, Pablo Bifani, Ghislain M. C. Bonamy, Tewis Bouwmeester, Clemens H. M. Kocken, Thierry T. Diagana | 2017 | Malaria Liver Stages Transcriptome | https://www.ncbi.nlm.nih.gov/sra/?term=SRP096160 | NCBI Sequence Read Archive, SRP096160 |

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
