## [Decision Letter]

Thank you for sending your article entitled "The Plasmodium Liver-Specific Protein 2 (LISP2) is an early marker of liver stage development" for peer review at *eLife*. Your article is being evaluated by three peer reviewers, and the evaluation has been overseen by a Reviewing Editor and Gisela Storz as the Senior Editor.

Extensive discussion between the reviewers and editors led to a list of essential revisions, including new experiments, given below.

Essential revisions:

1) In support of your claim, please address the question of antibody specificity experimentally to demonstrate that the protein is a marker for actively developing liver parasites. This experiment is needed as you only validated the specificity against recombinant antigen. While you provide IFA results, one suggestion would be to test the antibodies by western blots against cell lysates of infected hepatocytes.

2) In view of the significance of the LISP2 as an earlier marker of reactivation, you need to show that reactivating cells become LISP2 positive prior to the increased abundance of other potential protein markers of active growth and/or metabolism.

3) Normalize the staining intensity of H3K9 per nucleus.

4) Reanalyze the microscopic images/samples using z stacks by 3D reconstruction as it would clearly distinguish LIPS1 localization.

5) Revise the manuscript so that it reflects current status of *P. vivax* research.

6) Correct some factual information as indicated by the reviewers.

7) Include pertinent references such as Orito et al.

Reviewer #1:

This is a strong manuscript and was clearly presented although there are a number of minor typographical errors. The authors describe a protein, LISP2, that appears to be a reliable marker of hypnozoite reactivation in liver stage malaria parasites. This is an exciting discovery since two human malaria species (*P. vivax* and *P. ovale*) produce dormant hypnozoites that can reactivate days or months later, greatly complicating malaria treatment and diagnosis. Although 8-aminoquinolines like primaquine and tafenoquine are active against hypnozoites, drug discovery of new compounds and compound classes would be facilitated by the discovery of a molecular marker able to discriminate between hypnozoites and similarly sized cells that are in the early stages of reactivation. The authors describe the discovery of LISP2 in a transcriptomic study of liver stage *P. cynomolgi* malaria parasites (a simian model for *P. vivax*) and examine LISP2 expression by IFA and FISH in a primary simian hepatocyte culture system. These data and particularly the quantitative analysis shown in Figure 3B make an excellent case for hynozoites remaining LISP2 negative until reactivation and similar phenomena are shown for *P. vivax* parasites in an in vivo humanized mouse infection system. Consistent with these findings, the authors show that LISP2 negative *P. cynomolgi* parasites (presumably hypnozoites) are sensitive to tafenoquine, but not to other antimalarial drugs. The drug susceptibility data in particular highlight the utility of LISP2 as marker for hypnozoite reactivation. For this purpose, it would be even better if LISP2 was actually a maker of hypnozoites rather than a marker of non-dormant parasites, but the authors state in their Introduction that transcriptomic studies have so far failed to identify a marker for the hypnozoites themselves. There is one area where the authors do not do a good job of highlighting their discovery. I would assume that a whole host of genes are expressed at high levels in metabolically active liver stage parasites compared to dormant hypnozoites. The significance of the LISP2 discovery would be that it is an earlier marker of reactivation. Can the authors show that reactivating cells become LISP2 positive prior to the increased abundance of other potential protein markers of active growth (enzymes of processes like glycolysis or nucleotide metabolism)?

Reviewer #2:

This manuscript addresses the reactivation of dormant liver stages (hypnozoites) in the in vitro *P. cynomolgi* model as well as in in vivo (humanized mice) *P. vivax* model. Identification of markers distinguishing between arrested and developing liver stages is of great importance for the development of drugs targeting hypnozoites. Previous transcriptional profiling of small (early and dormant) and large (schizont) liver stages revealed that the lisp2 transcript is abundant in large stages and is also expressed in small stages. Remarkably, the lisp2 expression could distinguish three populations of *P. cynomolgi* liver stages, two of which were previously described – 1) small dormant hypnozoites, 2) developing multinucleated liver schizonts and an additional new population of 3) small liver trophozoites in the process of reactivation. Similar results were obtained in *P. vivax*-infected humanized mice. The notion that LISP2 marks parasites that are actively developing in the liver is further supported by the selective drug susceptibility of these stages. The manuscript is written clearly, presented data are of good quality and conclusions are valid. The authors should address some spelling mistakes within the manuscript text.

There are a few weak points that should be addressed:

1) While authors state that transcriptional activity is much weaker in dormant hypnozoites than in developing trophozoites, they only show H3K9Ac staining for hypnozoites and schizonts (not trophozoites) in Figure 3—figure supplement 2. This marker could be very useful to identify the point of reactivation (hypnozoite to trophozoite transition) and shed more light on whether the lisp2 expression indeed tightly correlates with this transition.

2) All LISP2 detection was performed using staining with a single antibody (two antibodies for *P. cynomolgi* and *P. vivax*, respectively). These antibodies were raised against recombinant proteins corresponding to a part of the LISP2 protein. Importantly, the specificity of the antibodies was verified by western blots using the same recombinant proteins. To make sure that the native protein is specifically recognized, lysates of infected liver cells should be used for verification. Given the high LISP2 abundance, its detection should be possible despite the low amount of parasite material in the infected liver cells lysate.

3) *P. berghei* LISP2 protein has been shown to be exported to the host cell cytoplasm and the nucleus (Morita et al., 2018), however, this does not seem to be the case for *P. cynomolgi* and *P. vivax* LISP2 in this study. Detection of LISP2 in infected liver cells by Western blot (as suggested in point 2) would additionally shed light on the processing of LISP2 in the liver stages of *P. cynomolgi* and allow to speculate on the causes of differential localization (while ruling out the option that the observed localization is an artefact based on cross-reactivity with other antigens).

4) Antibody recognizing PV1 is not well characterized. Western blotting using lysates of infected liver cells (see point 2) should be used to verify its specificity.

5) Authors speculate that LISP2 localizes to the parasitophorous vacuole or its extensions (cytoplasmic vacuoles) based on colocalization with PV1. However, the localization in most images resembles vesicles of the secretory pathway (which were also suggested for localization of both LISP2 in Morita et al., 2018 and PV1 (dense granules) in Chu, Lingelbach and Przyborski, 2011). Assuming that the antibody specifically stains LISP2 (see point 2), imaging of z-stacks followed by 3D reconstruction could shed more light on whether LISP2 truly localizes to the PV and its extensions or rather to isolated vesicles of the secretory pathway.

6) Adding references to figures in the conclusions would, in my opinion, increase the clarity of the text for the reader.

Reviewer #3:

LISP2 is a protein specifically expressed by plasmodial hepatic stages, required for the hepatic *P. berghei* merozoite formation. The manuscript entitled "The Plasmodium Liver-Specific Protein 2 (LISP2) is an early marker of liver stage development" shows that this protein is differentially expressed in uni-nucleated hypnozoites and growing hepatic trophozoites/schizonts, using two species of relapsing parasites (*P. vivax* and *P. cynomolgi*). Parasite stages were both defined/characterized morphologically and pharmacologically. This negative marker of hypnozoites and positive marker of growing liver stages can be used together with the parasite size and ploidy to discriminate these two populations, e.g., in anti-parasite/relapse drug-screening assays.

Introduction, first paragraph: *P. vivax* does not affect 2.5 billion people. Please revise.

Introduction, first paragraph: The argument that *P. vivax* research remains neglected does not seem to be justified based on the authors' rational. According to the authors, *P. vivax* malaria represents only ~4% of the global malaria cases (8.5 M/ 219 M), which roughly matches the 5% of global investment. Please revise or better justify.

Subsection “LISP2 is an early marker of parasite development in the liver”, first paragraph: Please cite the manuscript of Orito et al., 2013 after the mention of PbLISP2. The description of PbLISP2 localization and function is also missing.

Subsection “LISP2 is an early marker of parasite development in the liver”, last paragraph: The way the phrase is constructed gives the impression that relapsing-plasmodial species have more similar LISP2 amino acid sequence when compared to other non-relapsing species. This does not seem to be the case, since the LISP2 of *P. ovale*, another relapsing-plasmodial species, has only 55% of AA identity with Pv LISP2 6-cys domain, while *P. coatneyi* and *P. knowlesi*, two non-relapsing parasites, have 76 and 74% of identity, respectively. This stronger similarity is probably due to their closer genetic origin than relapsing-capacity. Please rectify.

Subsection “LISP2 is an early marker of parasite development in the liver”, last paragraph: Anti-Pv and Pc LISP2 antibodies were tested against the respective recombinant proteins used to generate them, but not against the respective parasites/ infected hepatocytes. Since the 6-cysteine domain is conserved among plasmodial proteins, it would be judicious to test the antibodies in a Western Blot using infected hepatocytes to assess the complexity of the antibody binding.

Subsection “LISP2 is an early marker of parasite development in the liver”, last paragraph: What is the peripheral membrane? The parasitophorous vacuole membrane? It is not clear in the figures if LISP2 is colocalizing with UIS4, apart from the inset of Figure 2B (prominence).

Subsections “Assessment of LISP2 expression in *P. vivax* liver stages”, “LISP2 expression differentiates dormant hypnozoites from developing liver stages”, last paragraph and Discussion, second paragraph: LISP2- Hz and LISP2+ Tz does not seem to have identical size according to the Figure 3B and scheme of Figure 5 (<8 days). Please apply a statistical analysis and clarify.

The authors state that "schizonts were remarkably more transcriptional active than hypnozoites" based on the number of H3K9ac structures. Since schizonts are multinucleated, a more informative way to strengthen this point would be to normalize the staining intensity of H3K9 per nucleus.

Subsection “LISP2 expression differentiates dormant hypnozoites from developing liver stages”, last paragraph: Please justify the fact that LISP2- Hz and LISP2+ Tz are identical in size. Additionally, remove the "only" in the phrase since these two populations can apparently be distinguished based on the parasite size (Days 3-6), and also by the intensity (ploidy) of nuclear staining (please see the parasite nuclei in Figure 1—figure supplement 3). Please compare the intensity of the nuclear staining of LISP2- Hz and LISP2+ Tz and correct the scheme of Figure 5 accordingly.

Discussion, fifth paragraph: No data presented in the manuscript allows inferring the essentiality of LISP2 in the liver-stage development. Please remove "essential" from the phrase.

Figure 1—figure supplement 2 legend: At day 15, the pattern of LISP2 staining in the merged panel resembles the host cytoplasmic staining published by Orito et al., 2013. It would be important to confirm if Pv/Pc LISP2 is also exported to the cytoplasm/nucleus of the infected host cell.

---

## [Author Response]

Essential revisions:1) In support of your claim, please address the question of antibody specificity experimentally to demonstrate that the protein is a marker for actively developing liver parasites. This experiment is needed as you only validated the specificity against recombinant antigen. While you provide IFA results, one suggestion would be to test the antibodies by western blots against cell lysates of infected hepatocytes.2) In view of the significance of the LISP2 as an earlier marker of reactivation, you need to show that reactivating cells become LISP2 positive prior to the increased abundance of other potential protein markers of active growth and/or metabolism.3) Normalize the staining intensity of H3K9 per nucleus.4) Reanalyze the microscopic images/samples using z stacks by 3D reconstruction as it would clearly distinguish LIPS1 localization.5) Revise the manuscript so that it reflects current status of P. vivax research.6) Correct some factual information as indicated by the reviewers.7) Include pertinent references such as Orito et al.Reviewer #1:This is a strong manuscript and was clearly presented although there are a number of minor typographical errors. The authors describe a protein, LISP2, that appears to be a reliable marker of hypnozoite reactivation in liver stage malaria parasites. This is an exciting discovery since two human malaria species (P. vivax and P. ovale) produce dormant hypnozoites that can reactivate days or months later, greatly complicating malaria treatment and diagnosis. Although 8-aminoquinolines like primaquine and tafenoquine are active against hypnozoites, drug discovery of new compounds and compound classes would be facilitated by the discovery of a molecular marker able to discriminate between hypnozoites and similarly sized cells that are in the early stages of reactivation. The authors describe the discovery of LISP2 in a transcriptomic study of liver stage P. cynomolgi malaria parasites (a simian model for P. vivax) and examine LISP2 expression by IFA and FISH in a primary simian hepatocyte culture system. These data and particularly the quantitative analysis shown in Figure 3B make an excellent case for hynozoites remaining LISP2 negative until reactivation and similar phenomena are shown for P. vivax parasites in an in vivo humanized mouse infection system. Consistent with these findings, the authors show that LISP2 negative P. cynomolgi parasites (presumably hypnozoites) are sensitive to tafenoquine, but not to other antimalarial drugs. The drug susceptibility data in particular highlight the utility of LISP2 as marker for hypnozoite reactivation. For this purpose, it would be even better if LISP2 was actually a maker of hypnozoites rather than a marker of non-dormant parasites, but the authors state in their Introduction that transcriptomic studies have so far failed to identify a marker for the hypnozoites themselves. There is one area where the authors do not do a good job of highlighting their discovery. I would assume that a whole host of genes are expressed at high levels in metabolically active liver stage parasites compared to dormant hypnozoites. The significance of the LISP2 discovery would be that it is an earlier marker of reactivation. Can the authors show that reactivating cells become LISP2 positive prior to the increased abundance of other potential protein markers of active growth (enzymes of processes like glycolysis or nucleotide metabolism)?

We agree with the reviewer that demonstrating that, LISP2 expression precedes the expression of a late liver schizont specific protein that is not expressed in the hypnozoite, would add value to our findings. We have tried to generate antibodies against such proteins with limited success so far and the PV1 antibodies reported here is the result of those efforts. The Ferredoxin antibodies we have previously reported (Voorberg et al., 2017) were made in mouse, like the LISP2 antibodies, which unfortunately precludes double staining IFA experiments. Nonetheless, we would like to point out that in these experiments we have not observed expression of Ferredoxin in the early developing liver-stage forms in an expression pattern that is consistent with the LISP2 expression we report here. Although we agree that only double staining could unambiguously prove this point, our data seem to suggest that indeed LISP2 expression precedes Ferredoxin protein expression in liver schizonts. Unfortunately, to our knowledge, there are no other available *P. cynomolgi* (or *P. vivax*) specific antibodies that would allow us to make such a statement more definitively. We will certainly try to address this question as soon as such reagents would become available.

Reviewer #2:This manuscript addresses the reactivation of dormant liver stages (hypnozoites) in the in vitro P. cynomolgi model as well as in in vivo (humanized mice) P. vivax model. Identification of markers distinguishing between arrested and developing liver stages is of great importance for the development of drugs targeting hypnozoites. Previous transcriptional profiling of small (early and dormant) and large (schizont) liver stages revealed that the lisp2 transcript is abundant in large stages and is also expressed in small stages. Remarkably, the lisp2 expression could distinguish three populations of P. cynomolgi liver stages, two of which were previously described – 1) small dormant hypnozoites, 2) developing multinucleated liver schizonts and an additional new population of 3) small liver trophozoites in the process of reactivation. Similar results were obtained in P. vivax-infected humanized mice. The notion that LISP2 marks parasites that are actively developing in the liver is further supported by the selective drug susceptibility of these stages. The manuscript is written clearly, presented data are of good quality and conclusions are valid. The authors should address some spelling mistakes within the manuscript text.

We believe we have addressed all major concerns raised by this reviewer including grammatical errors in the revised manuscript.

There are a few weak points that should be addressed:1) While authors state that transcriptional activity is much weaker in dormant hypnozoites than in developing trophozoites, they only show H3K9Ac staining for hypnozoites and schizonts (not trophozoites) in Figure 3—figure supplement 2. This marker could be very useful to identify the point of reactivation (hypnozoite to trophozoite transition) and shed more light on whether the lisp2 expression indeed tightly correlates with this transition.

We thank the reviewer for this useful request for clarification. We do not suggest that H3K9 allows for the quantitative assessment of transcriptional activity across liver-stages and our statement regarding the lower transcriptional activity of hypnozoites is solely based on our previously reported transcriptomic analysis (Voorberg et al., 2017 and Bertschi et al. 2018). We nonetheless carried out additional experiments and included H3K9ac and LISP2 co-staining for all liver-stages (i.e. Hz, Tz and Sz) so the reader can more directly appreciate the relative distribution and level of expression of H3K9ac and LISP2 throughout the development of the parasite. See revised Figure 3—figure supplement 2.

Additional reference:

Bertschi NL, et al. Transcriptomic analysis reveals reduced transcriptional activity in the malaria parasite Plasmodium cynomolgi during progression into dormancy. eLife 7, e41081 (2018).2) All LISP2 detection was performed using staining with a single antibody (two antibodies for P. cynomolgi and P. vivax, respectively). These antibodies were raised against recombinant proteins corresponding to a part of the LISP2 protein. Importantly, the specificity of the antibodies was verified by western blots using the same recombinant proteins. To make sure that the native protein is specifically recognized, lysates of infected liver cells should be used for verification. Given the high LISP2 abundance, its detection should be possible despite the low amount of parasite material in the infected liver cells lysate.

We assessed specificity of LISP2 in infected hepatocyte lysate and uninfected lysate by western blot. Since *P. cynomolgi* LISP2 is a high molecular weight protein (270Kda), day 8 infected samples were tested in two different lysis condition (denaturing and non-ionic cell lysis condition). In order to visualize the separation of high molecular weight LISP2, we subjected infected and uninfected lysates with Tris acetate SDS-PAGE and performed western blot (see Author response image 1). Under both denaturing and non-ionic lysis (data not shown) conditions, we were unable to detect any protein bands in both infected and uninfected samples. Similarly, we also did not obtain any protein band when we subjected lysates under Tris glycine SDS-PAGE. Additionally, we attempted to enrich the samples through immunoprecipitation experiments with LISP2 antibody prior to western blot analysis. However, we could not detect LISP2 protein or any other protein band in these experiments (data not shown).

We have included two positive controls for western and both worked well. We could detect the recombinant LISP2 epitope (37Kda) that was used to generate the anti-LISP2 antibody. In addition, we could also detect specific bands of the expected size for tubulin with anti-tubulin antibodies.

Given, the low infection rate (0.4%), we anticipated that, as mentioned by the reviewer, it would be difficult to detect LISP2 protein by western blot. However, the very low background observed in these experiments in uninfected/infected cells confirm that the anti-LISP2 antibodies are quite specific.

Moreover, RNA-FISH data with LISP2 probe (Figure 1—figure supplement 4) reveals the similar localization pattern observed with anti-LISP2 antibody in hypnozoites, trophozoites and schizonts. Taken together, the IFA data, RNA FISH data and western blot data corroborate the specificity of the anti-LISP2 antibody.

3) P. berghei LISP2 protein has been shown to be exported to the host cell cytoplasm and the nucleus (Morita et al., 2018), however, this does not seem to be the case for P. cynomolgi and P. vivax LISP2 in this study. Detection of LISP2 in infected liver cells by Western blot (as suggested in point 2) would additionally shed light on the processing of LISP2 in the liver stages of P. cynomolgi and allow to speculate on the causes of differential localization (while ruling out the option that the observed localization is an artefact based on cross-reactivity with other antigens).

We thank the reviewer for this question. We also investigated this previously and carried out extensive immunofluorescence microscopy from day 1 to day 21 post infection in vitro (Figure 1). We did not observe any LISP2 protein in hepatocytes at any time points. Additionally, we corroborated the absence of LISP2 protein in the host cell in a different relapsing model, *P. vivax* (Figure 2). Given the significant divergence in amino-acids sequences between *P. cynomolgi* and *P. berghei* LISP2 proteins, we cannot speculate whether LISP2 in *P. cynomolgi* is also proteolytically processed and we failed to detect LISP2 in western blot experiments.

4) Antibody recognizing PV1 is not well characterized. Western blotting using lysates of infected liver cells (see point 2) should be used to verify its specificity.

As mentioned above, due to low infection rate (0.4%), we could not detect LISP2 in infected and uninfected lysates. We also tried PV1 antibodies and likewise failed to see any specific bands.

5) Authors speculate that LISP2 localizes to the parasitophorous vacuole or its extensions (cytoplasmic vacuoles) based on colocalization with PV1. However, the localization in most images resembles vesicles of the secretory pathway (which were also suggested for localization of both LISP2 in Morita et al., 2018 and PV1 (dense granules) in Gupta et al.,2016). Assuming that the antibody specifically stains LISP2 (see point 2), imaging of z-stacks followed by 3D reconstruction could shed more light on whether LISP2 truly localizes to the PV and its extensions or rather to isolated vesicles of the secretory pathway.

To answer these questions, day 6 infected hepatocytes co-stained with PV1 and LISP2. 30-40 stacks of images were acquired with 100X objective lens with Z-step size of 1 μm. Three-Dimensional images were reconstructed using deconvoluted z-stack images. In the revised manuscript, we have incorporated 3-D videos of individual localization of nuclear stain, LISP2 and PV1 in mature schizonts.

Video 2 is the 3-D video of multinucleate schizont co-stained with LIPS2 and PV1.

PV1 is a known marker of parasitophorous vacuole. In blood stages, PV1 stains the parasite in circumferential pattern surrounding the parasite and described as ring of beads in Chu, Lingelbach and Przyborsk, 2011.

Our results recapitulate similar observations in hepatic stages and substantiate that LISP2 co-localizes with PV1 marker and stains the parasitophorous vacuolar membrane structures also shown in additional images in Figure 3—figure supplement 3.

Nonetheless, we do agree with the reviewer that it would be important to explore the vacuolar structures that resemble secretory vesicles. We will continue to investigate this question and to report our results as we are finding out more.

See Video 2, 3, 4 and 5 and Figure 3—figure supplement 3.

6) Adding references to figures in the conclusions would, in my opinion, increase the clarity of the text for the reader.

We incorporated those suggestions in the revised manuscript. See revised figure legends.

Reviewer #3:LISP2 is a protein specifically expressed by plasmodial hepatic stages, required for the hepatic P. berghei merozoite formation. The manuscript entitled "The Plasmodium Liver-Specific Protein 2 (LISP2) is an early marker of liver stage development" shows that this protein is differentially expressed in uni-nucleated hypnozoites and growing hepatic trophozoites/schizonts, using two species of relapsing parasites (P. vivax and P. cynomolgi). Parasite stages were both defined/characterized morphologically and pharmacologically. This negative marker of hypnozoites and positive marker of growing liver stages can be used together with the parasite size and ploidy to discriminate these two populations, e.g., in anti-parasite/relapse drug-screening assays.Introduction, first paragraph: P. vivax does not affect 2.5 billion people. Please revise.

We have changed the sentence as follows:

“*P. vivax* is the second most prevalent malarial pathogen, with a wider geographical distribution than *P. falciparum*, suggested to be a risk of malaria infection for 2.5 billion people.”

Introduction, first paragraph: The argument that P. vivax research remains neglected does not seem to be justified based on the authors' rational. According to the authors, P. vivax malaria represents only ~4% of the global malaria cases (8.5 M/ 219 M), which roughly matches the 5% of global investment. Please revise or better justify.

We thank the reviewer for this clarification and have thus revised the sentence as follows:

“Despite its high prevalence in many malaria endemic countries, *P. vivax* research is restricted to few laboratories and limited progress have been made.”

Subsection “LISP2 is an early marker of parasite development in the liver”, first paragraph: Please cite the manuscript of Orito et al., 2013 after the mention of PbLISP2. The description of PbLISP2 localization and function is also missing.

We have made the requested changes in the first paragraph of the subsection “LISP2 is an early marker of parasite development in the liver”.

Subsection “LISP2 is an early marker of parasite development in the liver”, last paragraph: The way the phrase is constructed gives the impression that relapsing-plasmodial species have more similar LISP2 amino acid sequence when compared to other non-relapsing species. This does not seem to be the case, since the LISP2 of P. ovale, another relapsing-plasmodial species, has only 55% of AA identity with Pv LISP2 6-cys domain, while P. coatneyi and P. knowlesi, two non-relapsing parasites, have 76 and 74% of identity, respectively. This stronger similarity is probably due to their closer genetic origin than relapsing-capacity. Please rectify.

We thank the reviewer for drawing our attention to this point.

We agree with that the 6-cys domain is highly conserved across *Plasmodium* orthologs and we have thus removed this sentence to avoid the possible confusion highlighted by the reviewer.

Subsection “LISP2 is an early marker of parasite development in the liver”, last paragraph: Anti-Pv and Pc LISP2 antibodies were tested against the respective recombinant proteins used to generate them, but not against the respective parasites/ infected hepatocytes. Since the 6-cysteine domain is conserved among plasmodial proteins, it would be judicious to test the antibodies in a Western Blot using infected hepatocytes to assess the complexity of the antibody binding.

See response to reviewer 2 point 2.

Subsection “LISP2 is an early marker of parasite development in the liver”, last paragraph: What is the peripheral membrane? The parasitophorous vacuole membrane? It is not clear in the figures if LISP2 is colocalizing with UIS4, apart from the inset of Figure 2B (prominence).

Here the peripheral membrane represents LISP2 staining around the parasite body in a circumferential manner. We later showed co-localization of LISP2 with PV1, a parasitophorous vacuole marker (Figure 3C).

We observed only partial overlap of LISP2 and UIS4 at the so-called prominence as seen in Figure 2B. In developing stages, these two proteins do not overlap as shown in Figure 2A.

Subsections “Assessment of LISP2 expression in P. vivax liver stages”, “LISP2 expression differentiates dormant hypnozoites from developing liver stages”, last paragraph and Discussion, second paragraph: LISP2- Hz and LISP2+ Tz does not seem to have identical size according to the Figure 3B and scheme of Figure 5 (<8days). Please apply a statistical analysis and clarify.

Here, we meant to say that these two parasite populations’ sizes range similarly. We apologies for this confusion. In the updated manuscript, we have amended the sentences accordingly to clarify this point (subsection “Assessment of LISP2 expression in *P. vivax* liver stages”, subsection “LISP2 expression differentiates dormant hypnozoites from developing liver stages”, last paragraph and Discussion, second paragraph).

The authors state that "schizonts were remarkably more transcriptional active than hypnozoites" based on the number of H3K9ac structures. Since schizonts are multinucleated, a more informative way to strengthen this point would be to normalize the staining intensity of H3K9 per nucleus.

As stated earlier in response to reviewer 2 (point 1), we do not suggest that H3K9 allows for the quantitative assessment of transcriptional activity across liver-stages and our statement regarding the lower transcriptional activity of hypnozoites is based on our previously reported transcriptomic analysis (Voorberg et al., 2017 and Bertschi et al. 2018). Thus we not believe that a normalization of the signal is warranted at this stage. Nevertheless, in the revised figure (Figure 3—figure supplement 2), we provided additional images of all three liver-stages (i.e. Hz, Tz and Sz) which allows for the visual resolution of individual nuclei so the reader can directly compare the distribution and expression of this marker across the development of the parasite in the liver. We also reconstructed 3-D images of schizonts using de-convoluted Z stack images co-stained with H3K9ac and LISP2. These Z stacks clearly showed LISP2 staining surrounding the cellular body in a circumferential pattern while H3K9ac localized to what appears to be individual nuclear structures within schizonts. In the revised manuscript, we have included this additional video (see Video 1).

Subsection “LISP2 expression differentiates dormant hypnozoites from developing liver stages”, last paragraph: Please justify the fact that LISP2- Hz and LISP2+ Tz are identical in size. Additionally, remove the "only" in the phrase since these two populations can apparently be distinguished based on the parasite size (Days 3-6), and also by the intensity (ploidy) of nuclear staining (please see the parasite nuclei in Figure 1—figure supplement 3). Please compare the intensity of the nuclear staining of LISP2- Hz and LISP2+ Tz and correct the scheme of Figure 5 accordingly.

See our seventh response to reviewer 3 (above). We have clarified throughout the similarity in size ranges throughout the manuscript and modified the sentence as suggested to remove the word “only”.

Unfortunately, as previously reported for *P. vivax* hypnozoites (March et al., 2013), DAPI staining does not allow for an accurate and quantitative assessment of ploidy.

We believe that with the above clarifications no changes to Figure 5 are warranted.

Discussion, fifth paragraph: No data presented in the manuscript allows inferring the essentiality of LISP2 in the liver-stage development. Please remove "essential" from the phrase.

We have updated the text as suggested (Discussion, fourth paragraph).

Figure 1—figure supplement 2 legend: At day 15, the pattern of LISP2 staining in the merged panel resembles the host cytoplasmic staining published by Orito et al., 2013. It would be important to confirm if Pv/Pc LISP2 is also exported to the cytoplasm/nucleus of the infected host cell.

We do not believe that we provided any evidence suggesting that the Pc LISP2 protein is exported to the host cell cytosol using the antibodies we raised against the 6-cys domain of the protein. We would also like to point out that we unsuccessfully attempted to raise antibodies against the Pc LISP2 N-terminal region which was shown to be exported to the hepatocyte cytosol in *P. berghei* parasites.

See also our response to reviewer 2 point 3.